environmental science/computer modelling and simulation

global trade, relation-driven trade, food security, micronutrients, agent-based model

**Author for correspondence:**
Jiaqi Ge
e-mail: j.ge@leeds.ac.uk

# Food and nutrition security under global trade: a relation-driven agent-based global trade model

Jiaqi Ge[1], J. Gareth Polhill[2], Jennie I. Macdiarmid[3], Nuala Fitton[4], Pete Smith[4], Heather Clark[5], Terry Dawson[6] and Mukta Aphale[4]

[1]School of Geography, University of Leeds, Leeds LS2 9JT, UK
[2]Information and Computational Science, The James Hutton Institute, UK
[3]The Rowett Institute, [4]School of Biological Sciences, and [5]The Institute of Applied Health Sciences, University of Aberdeen, Aberdeen AB24 3FX, UK
[6]Department of Geography, King's College London, UK

JG, 0000-0001-6491-3851; PS, 0000-0002-3784-1124

This paper addresses the highly relevant and timely issues of global trade and food security by developing an empirically grounded, relation-driven agent-based global trade model. Contrary to most price-driven trade models in the literature, the relation-driven agent-based global trade model focuses on the role of relational factors such as trust, familiarity, trade history and conflicts in countries' trade behaviour. Moreover, the global trade model is linked to a comprehensive nutrition formula to investigate the impact of trade on food and nutrition security, including macro and micronutrients. Preliminary results show that global trade improves the food and nutrition security of countries in Africa, Asia and Latin America. Trade also promotes a healthier and more balanced diet, as countries have access to an increased variety of food. The effect of trade in enhancing nutrition security, with an adequate supply of macro and micronutrients, is universal across nutrients and countries. As researchers call for a holistic and multifactorial approach to food security and climate change (Hammond and Dubé 2012 *Proc. Natl Acad. Sci. USA* **109**, 12 356–12 363. (doi:10.1073/pnas.0913003109)), the paper is one of the first to develop an integrated framework that consists of socio-economic, geopolitical, nutrition, environmental and agri-food systems to tackle these global challenges. Given the ongoing events of Brexit, the US–China trade war and the global COVID-19 pandemic, the paper will provide valuable insights on the role of trade in improving the food and nutrition security across countries.

# 1. Background

In 2019, 821.6 million people in the world are hungry and 2 billion people (26.4% of the world's population) are food and nutrition insecure, who lack access to safe, sufficient and nutritious food, according to the State of Food Security and Nutrition [1]. Despite great progress in agri-technology in the last century, hunger is increasing in poor countries, especially during economic slowdowns and downturns. In addition to the degradation of ecosystems and more frequent crop failures due to climate change [2], recent trade disputes and conflicts among countries, and the outbreak of the COVID-19 pandemic have all posed threat to global food supply and food security [3]. As a result, the world is struggling to meet the WHO nutrition target by 2025 or the UN sustainable development goal's nutrition target by 2030.

Food production and supply concerns more than the agri-food systems. They are deeply coupled with the social, economic and geopolitical systems at both local and global scales. Using a large-scale simulation model of land-use change, Brown et al. [4] show that social and behavioural factors can drastically change local land use and cause severe food shortages of up to 56% without climatic disturbances. Hammond and Dubé [5] argue that food and nutrition security is driven by complex underlying systems at both local and regional/global scales. The authors call for a systems approach using transdisciplinary modelling tools such as system dynamics and agent-based modelling.

One important mechanism for food allocation is via trade. In 2013, about 23% of the food produced for human consumption was traded internationally [6], which feeds 2–3 billion people globally and uses 13% of worldwide cropland and pasture [7,8] and the proportion continues to grow annually. As globalization deepens, spatially distant countries become increasingly connected by trade, and so are their agri-food systems, a phenomenon called 'telecoupling'. For example, Challies et al. [9] show how surging pork demand in Germany caused large-scale deforestation in Brazil, as the latter is a major supplier of soya beans, which are used as pig feed in Germany [9]. Similarly, Fuchs et al. [3] discuss how the recent US–China trade war can cause large-scale deforestation in the Amazon, as China switched to Brazil for soya imports. Disputes and conflicts between any two countries can thus lead to a serious disturbance to global trade and food security.

## 1.1. Price versus relation-driven trade models

Conventional trade models are based on the general equilibrium (GE) theory in economics, where trade volumes are expressed as mathematical equations of commodity prices. Solutions to the equations are found (analytically or numerically) by adjusting the prices so that supply equals demand (market clearing) in every sector. One of the common methods to derive numerical solutions is computable general equilibrium (CGE), which is used in models such as global trade analysis project (GTAP) [10]. CGE trade models have been used to evaluate the impact of trade policies such as tariffs and trade liberalization (e.g. [11]). Apart from GE models, partial equilibrium (PE) models are also developed for the agricultural sector, upon which integrated assessment models such as common agricultural policy regionalized impact model (CAPRI) [12] are based. Both CGE and PE models are built upon the key assumptions of equilibrium, price elasticities and the market-clearing condition. However, these theoretical frameworks offer limited possibilities for rigorous testing against historical data and experience [13].

The gravity model of trade, on the other hand, takes an empirical approach, which assumes that trade volume between two countries is proportional to the GDP of the two countries and disproportional to their geographic distance [14,15]. The model has since become one of the most successful empirical models in economics [16,17]. A price mechanism and market clearing, which are central in CGE and most macroeconomic trade theories, are not part of the gravity trade model. There is thus a gap between CGE model, which is based on theory but lacks predictive power, and the gravity model, which has predictive power but lacks theoretical underpinning.

What conventional price-driven trade models like CGE do not consider is other relational factors such as trust, familiarity, conflicts and competition in a trade relationship. For example, evidence shows that countries that share culture, language, religion and institutional structure trade more even after controlling for geographic distance and GDP [18–25]. Trust and familiarity play an important role in trade relationships. Using trade data of 25 countries, Den Butter and Mosch [26] show that trust explains a large fraction of trade volume between any two countries by reducing transaction costs. Deardorff [27] argues that increasing trust and reducing unfamiliarity is crucial in establishing new

trade relationships, especially for countries culturally high in uncertainty aversion. Huang [28] shows empirically that transport cost is not the only reason that distant countries trade less. Countries trade also less because they have never traded before and are thus unfamiliar with each other.

For perishable goods such as food and agricultural products, trust, reputation and previous trade relationship is particularly important. Because the conditions of perishable goods are hard to enforce by contract compared with manufactured goods, importers and exporters rely on mutual trust and repeated transactions to assure contractual performance. Macchiavello and Morjaria [29] show the value of long-term relationships based on trust and reputation in rose exports in Kenya. They find that when supply disturbances occur, exporters have incentives to prioritize delivery to buyers with whom they have traded before to protect their reputation. Similarly, research has found that trust, reputation and a previous trade relationship are particularly important for perishable and non-enforceable goods such as foods and agricultural products [30,31].

Trade can also be driven by geopolitical conflicts and competition among countries. For example, China's decision to buy soya from Brazil and not the US is driven by its relationship with the US rather than any 'rational' economic factors. Similarly, Brazil uses soya bean exports to secure land ownership in neighbouring countries, particularly Paraguay and Bolivia, extend political influence in Africa and balance trade with China [32]. Countries may also engage in (irrational) competition with each other for essential foods, especially in a crisis. Timmer [33] shows how, in the 2008 rice crises, countries in Southeast Asia competed to secure rice by hoarding and banning exports, causing spikes in rice price far exceeding what classic economic theory would suggest based on the initial (moderate) fall in supply. Timmer [34] concludes that complex human behaviours, such as loss aversion, time inconsistency and herd behaviour can present significant challenges to a traditional 'economic optimization' approach to trade.

In summary, existing price-driven global trade models are insufficient to fully capture trade relationships among countries. Relational factors such as trust, familiarity, conflicts and competition are just as important. When trust and a previous trade relationship is needed for trade to happen, countries will miss potential trade opportunities if they don't have trust between them [26], leaving the market to not clear in reality. We thus need a relation-driven approach to global trade to complement the price-driven one, which this paper will develop.

## 1.2. Countries as agents

Although trade activities are carried out by individual firms, most economic global trade models such as CGE and gravity models treat countries as the entity of trade. Similarly, many agent-based models (ABMs) of trade and international relations also treat countries as encapsulated agents that interact with each other [22,35,36]. In fact, the question 'what constitutes an agent' remains open in the ABM community. Macal and North [37] summarize four properties most 'agents' in ABM have: (i) autonomy (function independently in their environment), (ii) modularity (identifiable, discrete entity), (iii) sociality (interact with other agents), and (iv) conditionality (have a state that varies over time). Entities that meet the above four criteria have been represented as agents in the ABM literature, ranging from a biological cell to a person to an organization to a country.

Conceptually, it is appropriate to treat countries as agents if the model aims to study country-wise trade relationships, because a country makes decisions on national trade policies and positions, and behaves as a single entity as it sets up trade barriers, joins trade agreements and engages in trade wars [34,35]. Moreover, a country shares common attributes such as GDP per capita, institutional structure, languages and culture, so it can be treated as a single agent when researchers look at the role of these attributes in trade [22,23]. On the other hand, treating countries as a single unit misses the complex behaviour of and interactions between individual firms that carry out the actual trade. Treating countries as agents also miss the internal trade flows between regions in a country. Hence more other studies have been using firm-level and/or regional-level data to study heterogeneous firm behaviour in trade (e.g. [29]) and intra-national trade (e.g. [38]).

Practically, however, most global trade data are available at the country level, including the ones from the UN and OECD, which we use in this study. Although some advanced countries have much more detailed trade data at the regional level, such segregated data are not available for all countries in the world, especially the lower income ones that are more prone to food insecurity. Even more scarce are firm-level transaction data, which are often exclusive, almost always incomplete and vary greatly by countries and sectors. The lack and incompleteness of empirical data at lower than the country level severely restricts the types of empirically grounded agent-based model researchers can develop,

especially if the model aims to include all countries and sectors. Finally, as Ge and Polhill [39] have argued, agent representation is rather a narrative concept, and the appropriate agent will depend on the aims and purposes of the study, the research questions, and the constraints imposed by data, ontological complexity and computational capacity.

## 1.3. Multivariate nutrition

Most research on food security has so far focused on energy consumption (calories). However, having sufficient energy does not guarantee a nutritionally adequate diet. While obesity becomes a problem even in the poorest parts of the world [40], nutrient deficiency, especially deficiency in micronutrients, are still prevalent in both low- and high-income countries [41]. For example, more than 2 billion people in the world are deficient in iron; 21% of children are deficient in vitamin A, which is the direct cause of 800 000 deaths per year [42]. In 1996, the Food and Agriculture Organization (FAO) amplified the definition of food security to include a sufficient supply of nutrients in the diet. The UN's sustainable development goal (SDG Target 2) 'Zero Hunger' aims to meet the nutritional needs of all people, especially those susceptible to micronutrient deficiency (adolescent girls, pregnant and lactating women and older persons).

Previously the majority of food security research has focused on staple foods that are the main sources of calories, such as wheat, rice, soya beans and maize (e.g. [43–45]), although more recent studies have been looking at a larger variety of foods (e.g. [46–48]). To achieve food and nutrition security by FAO and SDG 2 standards, however, one needs a diverse, balanced diet containing a variety of foods, such as those rich in vitamin A (offal, oranges, carrots), iron (e.g. red meat, offal, spinach) and zinc (e.g. meat, seafood, nuts).

This paper develops an empirically grounded, relation-driven agent-based global trade model to study the impact of global trade on food and nutrition security of countries across the world. It addresses the highly relevant and timely issues of trade and food security given the current debates about trade agreements associated with Brexit and US–China trade dispute. The paper makes several important contributions to the current literature. First, it will develop a relation-driven global trade model to complement the price-driven trade models that dominate the literature. It will provide a more flexible framework to incorporate more complex, relation-driven trade behaviour of countries. The model developed in this paper can thus enhance our understanding of some trade phenomenon such as repeated trade, preferential treatment in trade and trade conflicts, which are common in the real world but hard to explain using price-driven trade theories. It also relaxes the restrictive assumptions in CGE models, including market-clearing and equilibrium condition and price-driven trade behaviour. Second, the trade model will include a comprehensive list of foods and be linked to a nutrition formula based on food consumption to investigate the impact of trade on food and nutrition security. The model can thus be used to identify countries and regions most vulnerable to food and nutrition shortage, and which macro- and micro-nutrition they are likely to lack under different scenarios.

# 2. Methods

This paper describes the development of an empirical agent-based model of global food trade to study the impact of trade and climate change on food and nutrition security of countries across the world. The empirical agent-based model is implemented in NetLogo [49]. In this section, we will follow the guidelines of Grimm *et al.*'s overview, design concepts and details (ODD) protocol [50,51] to describe the model.

## 2.1. Purpose

The purpose of the model is to study the impact of global food trade on food and nutrition security in countries around the world. It will incorporate three main aspects of trade between countries, including a country's wealth, geographic location and its trade relationships with other countries (past and ongoing), and will be used to study food and nutrition security across countries in various scenarios, such as climate change, sustainable intensification, waste reduction and dietary change.

## 2.2. Agent classes and attributes

### 2.2.1. Countries

As previously discussed, for both conceptual and practical reasons, we choose to represent countries as agents, which we believe is the most suitable for the study and the research questions we will address, as well as the most practical given the data and computational constraints. Each country in the model has a list of attributes (table 1), including geographic location, population size, GDP, production of (multi-dimensional) food commodities, the country's historic trade relationships with other countries and so on. We include the 165 countries in the world for which complete data of the food supply from the FAO food balance sheet are available. The model is spatially explicit in that the countries are represented spatially at a global scale.

The activities country agents engage in are the production, trade and food intake, which is a multi-dimensional variable consisting of 91 food commodities, consistent with those used in FAO food balance sheets (FBS) (see §2.4.3). The production of food commodities in each country is exogenous and changes every year depending on the scenarios. Once the production for the year is revealed or completed, countries trade with each other if they have unfulfilled domestic demand or unconsumed domestic supply. A country's food supply is then a combined outcome of its domestic production and trade with other countries. We do not consider inequalities within a country in access to food, which can be great in some countries. In this study, we focus on the average food intake per capita, which we use to compare with food requirement per capita, as an indicator of a country's food and nutrition security. Indicators of country-specific inequality (such as the GINI index) can be built into the model later on, which is beyond the scope of this study.

Table 1 lists selected attributes of a country agent.

#### 2.2.1.1. Trade intermediary

For a given commodity, an intermediary country is one that imports food for the purpose of re-exporting. As discussed before, intermediary countries are important facilitators of trade. In the model, we define a country as an intermediary for the commodity if the total export of the commodity is more than 80% of the total import (i.e. the majority of import is for re-exporting) in 2000, the baseline year. The motivation and trade behaviour of an intermediary country will be different from other non-intermediary countries that import for domestic consumption.

#### 2.2.1.2. Typical diet

We use the term 'diet' in the paper, but this is based on the national supply of food taken from the FAO food balance sheets (FBS), adjusted by the proportion that is inedible (e.g. banana peels) and wasted, which differs by region. Typical diet varies across countries and reflects a country's tradition and culture, as well as their natural and land-use conditions. When considering nutrient sufficiency and dietary change, we need to make sure that we do not naively prescribe countries an 'ideal diet' (nutritionally adequate) that is unrealistic to implement. In the model, we use the average reported food consumption between 2000 and 2002 (to smooth out fluctuations in any one year) as the baseline for each country's typical diet. The food composition in the typical diet changes every year in proportion to global food production and supply, to reflect the fact that the diet of people changes gradually (not drastically) over time. We assume that a country will aim to obtain the typical diet for its population in the current year; it will import a food commodity if it produces less domestically than is needed in the typical diet and export if it produces more. Some countries may fail to feed their populations with the typical diet; nor does a country's typical diet necessarily guarantee nutrient sufficiency, which reflects the situations in reality.

#### 2.2.1.3. Fortification

In some countries where wheat is refined and stripped of fibre and micronutrients, the flour and refined cereals are fortified replacing some of the micronutrients. The most commonly fortified food is wheat (flour) [53]. Hence, when calculating nutrient supply for each country, we need to adjust for the nutrient content of wheat in each country depending on if it is refined or not, then if the refined flour is fortified. This varies by the income of countries, with higher income countries more commonly refining than fortifying with micronutrients. The level of fortification also varies by country. The refined flour and

**Table 1.** Selected attributes of a country agent.

| variable | description | data source (if exogenous) | En?[a] | D?[b] |
|---|---|---|---|---|
| country name | name of the country, including variations | FAO | N | N |
| location | country location in a global map | GIS | N | N |
| area | geographic area | GIS | N | N |
| region and sub-region | the region and sub-region a country belongs to | UN | N | N |
| initial GDP | In 2000 | FAO | N | N |
| initial population | In 2000 | FAO | N | N |
| initial production | tonnes of each food commodity produced in the country in 2000 | FBS | N | N |
| initial import | tonnes of each commodity imported from all other countries in 2000 | FBS | N | N |
| initial export | tonnes of each commodity exported from this country to all other countries in 2000 | FBS | N | N |
| initial domestic supply (DS) | DS of all commodities in the country in 2000 (DS = production + import − export − stock = food + feed + processing + loss) | FBS | N | N |
| percentage of food in DS | percentage of food in DS for each commodity in 2000 (food/DS) | derived from FBS | N | N |
| percentage of loss in DS | percentage of loss in DS for each commodity in 2000 (loss/DS) | derived from FBS | N | N |
| is intermediary[c] | whether a country is a trade intermediary for each commodity | derived from FBS | N | N |
| import needed—domestic | import needed for domestic consumption for each commodity | endogenous | Y | Y |
| import needed—intermediary | import needed for re-export for each commodity (only relevant for intermediaries for the commodity) | endogenous | Y | Y |
| export available | export available for each commodity | endogenous | Y | Y |
| import realized | import (both domestic and intermediary) realized in each commodity | endogenous | Y | Y |
| export realized | export (both domestic and intermediary) realized in each commodity | endogenous | Y | Y |
| current food supply | food supply of each commodity in the current year | endogenous | Y | Y |
| current GDP | GDP in the current year | FAO if year $\leq$ 2013; OECD projection if year >2013 | | |
| current population | population in the current year | FAO if year $\leq$ 2013; OECD projection if year > 2013 | N | Y |
| current production | production of each commodity in the current year | FBS if year $\leq$ 2013; Scenario projection if year > 2013 | N | Y |

(*Continued.*)

| variable | description | data source (if exogenous) | En?[a] | D?[b] |
|---|---|---|---|---|
| typical diet[c] | the country's food supply of each commodity in 2000 | FBS | Y | Y |
| fortification | how the country fortifies its wheat products | food fortification initiative http://www.ffinetwork.org/index.html | N | N |
| household waste[c] | the percentage of household food wasted for each commodity | Gustavsson *et al.* [52] | N | N |
| population-level nutrient required per person[c] | population-level nutrient required per person in the given year in macro and micronutrients based on the demographic composition | WHO recommendations | N | Y |
| current nutrient supply consumed per person | average nutrient supply per person in the given year in macro and micronutrient | endogenous, calculated from food consumption | Y | Y |

[a]If the variable is endogenous (same for all tables in this section).
[b]If the variable is dynamic or will change over time (same for all tables in this section).
[c]More explanations below.

cereal, however, will have a lower supply of fibre. Food composition values are derived from the USDA food composition database (2014, release 27) [54]. Food Fortification Initiative specifies whether food is fortified or not in a country, and the income levels for countries are as specified by the World Bank.[1]

### 2.2.1.4. Household waste

Food waste up to the point of the household is accounted for in the FBS, but not waste generated in the household, after production and trade where a certain percentage of food will be wasted. Not accounting for household waste will lead to an overestimation of nutritional intake based on food consumption. The amount of food wasted depends on the type of food and the countries and regions. Generally speaking, countries and regions that are wealthier waste more food at the household level. Note that household waste does not include the part of food that is inedible, such as banana peel, which has already been accounted for in the nutrient calculation. Table 2 shows the percentage of food waste, based on food that could have been eaten, in household consumption by region and food type, which is estimated in Gustavsson *et al.* [52].

### 2.2.1.5. Nutrient requirements

Because people in different gender–age groups have different nutritional needs, the population-level nutrient requirement per person in a country is based on the demographic composition of its population. Countries with a larger young adult and male population will have a higher nutrient requirement than those with an ageing population. The demographic composition of a country will change over time, and so will the population-level nutrient requirement. An adequate energy intake was estimated using population-weighted average dietary energy requirements (ADERs), calculated using data for each age and sex with assumptions of physical activity level (PAL) being 1.75 and BMI being 21 kg m$^{-2}$ for adults. The population-level nutrient requirement of a country will be compared with the nutrient supply in the country in any given year to determine a country's level of nutrient sufficiency. The nutrient requirements are from the WHO [55,56].

[1]https://www.worldbank.org/

**Table 2.** Percentage of food waste in household consumption.

| region | cereals | roots & tubers | oilseeds | pulses | nuts | fruit | veg | meat | offal | fish | milk |
|---|---|---|---|---|---|---|---|---|---|---|---|
| Europe | 25 | 17 | 4 | 4 | 4 | 19 | 19 | 11 | 11 | 11 | 7 |
| North America, Oceania | 27 | 30 | 4 | 4 | 4 | 28 | 28 | 11 | 11 | 33 | 15 |
| industrialized Asia | 20 | 10 | 4 | 4 | 4 | 15 | 15 | 8 | 8 | 8 | 5 |
| sub-Saharan Africa | 1 | 2 | 1 | 1 | 1 | 5 | 5 | 2 | 2 | 2 | 0.1 |
| North Africa, West and Central Asia | 12 | 6 | 2 | 2 | 2 | 12 | 12 | 8 | 8 | 4 | 2 |
| South and Southeast Asia | 3 | 3 | 1 | 1 | 1 | 7 | 7 | 4 | 4 | 2 | 1 |
| Latin America | 10 | 4 | 2 | 2 | 2 | 10 | 10 | 6 | 6 | 4 | 4 |

### 2.2.2. Trade relationships between countries

As previously said, the trade model will be relation-driven, which means that trade decisions will depend on previous bilateral trade relationships between countries as they engage in trade repeatedly over time, in addition to their geographic location and ability to pay. We assume countries rank each other with different trade priorities to determine whom to trade with (assuming there are multiple competing buyers or sellers). Trade priority will depend on four elements: (i) GDP per capita, (ii) geographic distance, (iii) historic trade relationship, and (iv) emergent trade relationship.

The first element, GDP per capita, serves as a proxy for a country's ability to pay for a commodity. Priority is given to countries with a higher GDP per capita or high ability to pay. The second element, geographic distance between the two trading countries, is an important factor in predicting trade volumes: countries close to each other tend to trade more. One reason is the lower transport cost. Another reason is that countries close to each other are also more familiar with each other, and more likely to have a similar culture, customs and languages, all of which facilitate trade [28].

The third element, historic trade relationship, is measured by trade volume of all commodities in the year 2000. We use trade volume as an indicator of the existing trade relationships established between the two countries. As research has shown, countries that are more familiar with each other (via common language, religion, institutional structure and other social and cultural characteristics) trade more often [19,20,57]. While geographic proximity is one cause for enhanced familiarity and trust between countries, there are other non-geographic factors, such as historic connections (e.g. former colony, commonwealth) [58] and international organizations or trade unions (e.g. OECD, EEA, Trans-Pacific Partnership), that could cause some countries to have closer connections and thus trade more. Historic trade volume will reflect these non-GDP and non-geographic factors. We distinguish trade volumes by exports and imports because they represent different roles in trade relationships.

The last element, emergent trade relationship, allows new trade to emerge endogenously and to influence subsequent trade development in a path-dependent way. While the first three elements are exogenous to the model and deterministic, the last element is endogenous and stochastic. Trade relationship that evolved from the model has an impact on future trade decisions. Two countries low on each other's trade priority (due to low GDP per capita, long geographic distance or few trade records before) can start a new trade relationship in a year when they fail to trade with their usual partners (e.g. due to crop failure or market disturbances). Such an 'incidental' new trade between the two countries will increase their ranks in each other's trade priority, which will in turn increase the chances that they trade again in the future.

The trade priority (for importing and exporting partners) assigned by countries to one another is a weighted average of the four elements above. The weights are calibrated using actual trade and consumption data. We allow the weight to be zero in the search space, so that if any of the above elements do not have a significant impact on trade patterns in the empirical data, it will not have an effect in the model either. Although the second and third elements may be correlated (e.g. many countries that trade often are also geographically close), they do not coincide. For example, countries that have developed historical trade links may not be geographically close, such as the commonwealth countries; on the other hand, geographically close countries may not trade as much, such as the US and Cuba. Hence, we should still be able to distinguish the effect of the four elements when calibrating the weights for the elements using empirical data.

In this model, we do not try to emulate price dynamics or predict future prices. Although prices are not explicitly modelled, the price mechanism to allocate commodities among countries will be partially incorporated in the ranking and matching process of trade partners through countries' GDP per capita, geographic location, production, consumption and dietary preferences. A country's GDP per capita and dietary preference determine its purchasing power and willingness to pay for a product. Moreover, a country's production reflects its overall productivity and production costs of the food; its consumption reveals a country's budget, dietary habits and preferences; its location is a proxy for transportation costs. Therefore, although the model does not include the price mechanism directly, it does implicitly incorporate the information that prices contain.

Table 3 shows the attributes of the trade relationship class.

### 2.2.3. Food and nutrients

As was discussed before, we must look at a more comprehensive food list than a few major crops to gain a better understanding of the nutrient sufficiency, especially micronutrient sufficiency across countries. In

**Table 3.** Attributes of a trade relationship.

| variable | data source if not endogenous | En?[a] | D?[b] |
|---|---|---|---|
| GDP per capita | FAO (year <=2013); OECD projection (year > 2013) | N | Y |
| distance | GIS | N | N |
| historic trade relationship—import | FAO FBS data for years 2000–2002 | N | N |
| historic trade relationship—export | FAO FBS data for years 2000–2002 | N | N |
| emergent trade relationship—import | | Y | Y |
| emergent trade relationship—import | | Y | Y |

[a]If the variable is endogenous (same for all tables in this section).

[b]If the variable is dynamic or will change over time (same for all tables in this section).

this study, we include 91 food categories as in FAO's FBS. Countries will produce, trade and consume a different amount in each food category, which we then use to calculate macro and micronutrients. The aggregated groups to which each food category belongs are as follows: 'Cereals – Excluding Beer', 'Starchy Roots', 'Sugar Crops', 'Sugar & Sweeteners', 'Pulses', 'Treenuts', 'Oilcrops', 'Vegetable Oils', 'Vegetables', 'Fruits – Excluding Wine', 'Spices', 'Stimulants', 'Alcoholic Beverages', 'Meat', 'Offal', 'Animal fats', 'Eggs', 'Milk – Excluding Butter', 'Fish; Seafood', 'Aquatic Products' and 'Other (e.g. Infant food and miscellaneous)'.

The mapping of food to nutrients is based on data from the GENuS project [59]. For a detailed description of the methodology see [46]. Each of the 91 food categories, which are made up of many food items, was disaggregated to individual food items. Each food commodity was then mapped to the nutrient composition. A weighted mean of the nutrient composition for the food items was used when the data were aggregated back to the food commodity groups. This was based on the global production of each food item within that group (FAO production data). If there were no production data for food items in a food commodity, an unweighted average was used. The nutrient data came primarily from the USDA food composition data.[2] If any food item was not in the USDA tables, the composition was taken from other regional food composition tables. In this study, we exclude the nutrients for which the mapping involves large uncertainties, leaving the following: calories (energy, kcal), protein, fat, vitamin C, vitamin A, folate, calcium, iron, zinc, dietary fibre, thiamin, riboflavin, niacin, vitamin B6 and saturated fat.

## 2.3. Process overview and scheduling

Each step in the simulation represents a year in real time. We choose an annual time step because it is appropriate for the production cycle and the time-scale of the model (2000–2050), and also because data on trade, production, GDP and population are only available annually. The starting year of the model was 2000, which is also the baseline year.

At the beginning of each year, countries receive the production of all food from last year, which is the amount available for domestic consumption and trade. If a country has produced more than it needs for domestic consumption of the typical diet, it will try to export the excess; if a country has produced less than it needs for domestic consumption, it will try to import the deficiency. Also, if a country is an intermediary of a food commodity, it will want to import food for re-export, even though it may have a sufficient amount for domestic consumption.

After countries decide on their trade positions (whether to import or export, and the amount to import or export), they will start looking for trade partners. Importers will first send buying offers to potential exporters. If there are multiple exporters of the food on the market, importers will send the offer to the country with the highest trade priority on their lists (see §2.2.2). After importers have sent offers, exporters will decide to whom they will sell their commodity, if they receive multiple buying offers. Similarly, exporters will evaluate importers' priorities and sell only to the ones with the highest priority until all the available food for export is sold. Importers whose buying offers are not satisfied in this round will repeat the same process in the next round until their import demand is satisfied or there is nothing left to buy. To prevent some large countries with high demand from flooding and

---

[2]US Department of Agriculture, Agricultural Research Service. National Nutrient Database for Standard Reference, Release 27, 2014. Available online: http://www.ars.usda.gov/ba/bhnrc/ndl [54].

for food category *i*

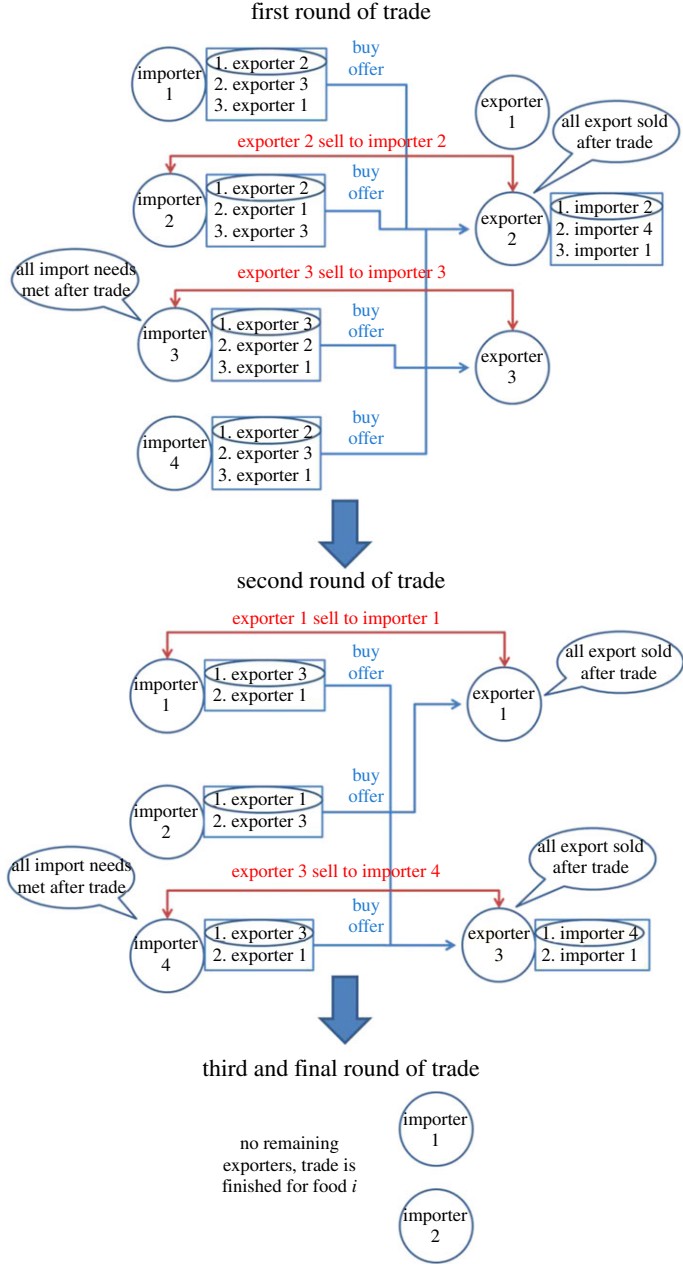

**Figure 1.** The trade procedure: the ranking and matching of importers and exporters.

dominating the market, we limit each trade transaction to a maximum amount, so that smaller countries can compete with larger countries. We emphasize that it is not the maximum amount a country can trade in each commodity in total; it is the maximum amount a country can trade in each commodity *per round*, and the trade of a commodity can take several rounds (on average 5–20) to finish (figure 1). Hence, countries with a large demand will try to buy again in subsequent rounds until their demand is fulfilled. We have conducted a sensitivity analysis using different values for the maximum amount per round, and the results are not sensitive to the parameter.

It will take several rounds for the market to clear and all the trade opportunities are realized, depending on the total trade volume and the matching process. The same process will then repeat for the next food commodity until all food categories have been traded. For simplicity, we assume that the trading outcome of one food category does not affect that of the other. Figure 1 illustrates the ranking and matching process of trade among countries.

One important factor in the model is trade saturation, which is the rate between realized and maximum trade amounts. Trade saturation ranges from 0% to 100%, which we vary in the computer

experiments. When trade saturation is 100%, all trade opportunities are realized. Countries will keep trading until either there are no more importers with remaining demand or exporters with remaining supply. There will be no missed trading opportunities, which can be regarded as an ideal case. When trade saturation is less than 100%, not all potential trade opportunities are realized. Countries at the bottom of the trade priority may not be able to trade, even though there are outstanding buyers and sellers of the same commodity. This 'less than ideal' case may be closer to reality, as factors such as to risk avoidance, market failure, lack of time (for perishable food), lack of information, communication or facilitators can prevent potential trade opportunities from being realized. The higher the trade saturation, the higher percentage of trade opportunities are realized. Later we examine the impact of trade saturation on food and nutrient security across countries.

## 2.4. Design concepts

### 2.4.1. Interaction

The countries interact with each other via trade. Links and connections are created between countries when one offers to buy from or sell to another. A country's decision to trade with another does not only affect the two countries involved in the trade, but it also indirectly affects other countries because once the food commodity has been traded, there will be less available on the market. Countries also interact with each other via their rankings of each other on trade priority. Their past interactions (trade) with other countries will influence the ranking as familiarity increases, which will in turn affect the way they interact/trade with others in the future.

### 2.4.2. Emergence

Trade relationships can emerge through repeated interactions among countries. Countries who have never traded before (and thus are low on each other's trade priority) can start a trade relationship by chance, for example, due to the lack of available trade partners at the time. The trade experience will then encourage the countries to trade more with each other in the future, thus the emergence of new trade relationships. The nutrients available per capita in each country emerge from the trade.

### 2.4.3. Stochasticity

The main source of randomness in the model comes from the matching of trade partners: the sequence of the countries matters when they send each other a buying offer. The random sequence of agents to execute functions is internal in NetLogo. To account for that internal stochasticity, we run the model 30 times for each parameter combination and look at the variance in results. As we can see in tables 15 to 21 in the Appendix, the variances caused by the internal stochasticity are very small. Apart from that, the model is data-driven and has not drawn other random parameters from distributions.

## 2.5. Initialization and data

The initialization of the model is based on empirical data or estimates from existing research. The 165 countries are created with initial GDP, population, production, food consumption using the average value of the years 2000, 2001 and 2002 in FAO's FBS data to smooth out any anomalies in any particular year. The past trade relationship is derived from FAO's trade data of all commodities. The geographic location and area of countries are initialized using a GIS world map. The regions and sub-regions to which countries belong are according to UN categorization and consistent with Müller *et al.* [44]. The typical diet for each country is initialized as the food consumption in 2000 from FBS. The income categories and the corresponding fortification type is derived from The World Bank and Food Fortification Initiative, respectively. The avoidable waste rate is initialized using estimates from estimation in Gustavsson *et al.* [52].

## 2.6. Input data

The current model runs from 2000 to 2013. Each year is input from data from FAO to update the GDP, population and production for all countries. The FAO FBS data is available until 2013. Table 4 shows the

**Table 4.** Input data for model running from 2000 to 2013.

| variable | data source | years |
|---|---|---|
| GDP | UN GDP of countries | 2000–2013 |
| population | UN GDP of countries | 2000–2013 |
| production (multivariate) | FAO food balance sheet: production | 2000–2013 |

input data for model running from 2000 to 2013. For scenario analysis in the future, the model will use projected data for production, GDP and population between 2014 and 2050.

# 3. Model calibration

## 3.1. Calibration

The parameters to be calibrated are the weights given to each of the four elements (GDP per capita, distance, historic and emergent trade relationships) in the countries' evaluation of trade partner priorities, which cannot be observed in empirical data. Because the ranking of countries is relative, we lose one degree of freedom, and hence we fix the weight for distance at 1 and allow the other three parameters to vary. The sampling of the parameters is twofold: first, we draw a sample of 10 000 random parameters from a three-dimensional Latin hypercube sampling ranging between 0 and 1; second, we transform the random sample using an exponential transformation to account for the ratio relationships (i.e. relative importance) between the weights, so that the parameters range from 0.05 to 20. In other words, relative to distance (which has a fixed weight 1), the weight for the other parameters range from 0.05 (1/20 as important) to 20 (20 times as important).

We then run the model on the 10 000 sampled parameter combinations. The empirical data we use for calibration and validation is FAO FBS on import and export volume and food consumption, and trade data from the United Nation Comtrade Database.[3] The data for validation and calibration is available between 2001 and 2013 (the model is initialized in 2000), of which the first seven years (2001–2007) is used for calibration, and the latter six years (2008–2013) for validation.

The evaluation of the model results is based on two dimensions: trade volume and trade partners. Trade volume compares the actual import, export and food consumption (in FAO FBS) in each country with the simulated results from the model. Because the simulated figures will almost surely not be the same as the observed ones (factors unaccounted for in the model, factors unobserved, errors in model specification, errors in input data, errors in empirical/validation data), we use a threshold to determine 'match': if the simulated volume is within ±20% of the actual volume (of import, export and food in each country), it is regarded as a match. The first dimension, trade volume, measures the percentage of simulated trade volumes (import, export, food) that falls within the range of the actual volume between 2001 and 2007.

The second dimension, trade partners, compares the actual bilateral trade between two countries in each food category with the simulated ones. If for a specific food category, the importer and exporter in the empirical data match that in the simulated data (regardless of trade volume), it is marked as a match. The trade partner index is the percentage of simulated trade records that matches the actual one of all trade records between 2001 and 2007. Because the FAO trade data does not contain information on bilateral trade partners or is not commodity-specific, we use the Comtrade data from the United Nations to calibrate and validate the bilateral trade results. The Comtrade data, however, does not cover all food categories in FAO. Of the 91 food categories, 42 are available in the Comtrade data.

To account for internal stochasticity, we run the model 30 times for each parameter combination and calculate the average matching rate. Figure 2 shows the 10 000 parameter combinations (consisting of three weights) plotted by the two evaluation dimensions: average matching rate for trade volume and trade partners. The higher the matching rate in each dimension, the better the model with that parameter combination performs. Tables 13 and 14 in Appendix show a small subset or example of the simulated trade volume and partners from the calibrated model (versus actual data), which we

[3]https://comtrade.un.org/.

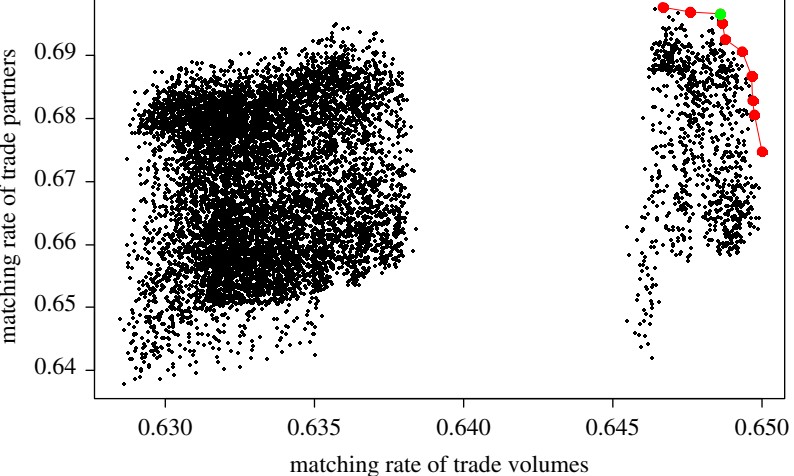

**Figure 2.** The Pareto front of the 10 000 parameter combinations.

**Table 5.** Calibrated weights for GDP, distance, historic and emergent trade in trade priority evaluation.

| GDP | distance (fixed) | historic trade | emergent trade |
|---|---|---|---|
| 1.40 | 1 | 1.40 | 0.72 |

use to calculate the matching rate. The full simulation results for the calibrated model are available in the electronic supplementary material, due to the large size of the data.

Figure 2 shows the Pareto front (the red line connecting the points) of the parameter combinations. The Pareto front represents the set of optimal parameters in that the model performance cannot be improved on one dimension without lowering the performance on the other dimension. There are 10 points on the Pareto front, as shown (in red and green) in figure 2. There is a big gap in the graph, which shows that a group of parameter combinations (those to the right of the gap) does significantly better in matching trade volumes than the rest. The data used to generate the graph, i.e. the mean match rate of trade partners and trade volumes for all 10 000 parameter combinations is available in the electronic supplementary material, data.

By definition, all the points on the Pareto front are incomparable and can only be partially ordered. Since we value the two performance dimensions equally, we would like to choose a parameter combination that does relatively well in both dimensions. We then calculate the average of the two dimensions (i.e. give equal weight to each) and pick the point with the highest average (in green in figure 2), which will be the parameters for the calibrated ABM. The calibrated values of the weight for GDP, distance (fixed at 1 as the bench market), historic and emergent trade (in trade priority evaluation) are listed in table 5. We see that GDP and Historic trade have higher weights than distance and emergent trade, indicating a relatively bigger influence in countries' ranking of trade partners. However, the other two elements are also significantly different from zero, which means they also play a role. There are two caveats concerning the validation method. First, the parameters are the same for all countries (lack of parametrization per country). Second, there are potential although not linear correlations between the four elements in the ranking of trade partners as discussed in §2.2.2. However, they are not an issue for the study, which includes the prediction trade volumes among countries based on their trade relationships.

## 3.2. Discussion on calibration and validation

To save space, the validation results are presented in appendix A.1. Overall, we find that the model has a better predictive power for trade partners than trade volumes, which is unsurprising partly because the calibrated parameters are in the functions of countries to rank and match trade partners. Moreover, the errors in input data for the volume of trade are likely to be larger than that for trade partners. As a result, we expect the validation results for volume to be less accurate than that for trade partners. We also find

that the predictive power of the model decreases over time from 2008 to 2013. As the model moves away from the original year it is initialized (2000), and the years on which is it calibrated (2001–2007), we expect its predictive power to go down, and the variances among parallel models to go up as the effects of stochasticity and random events from previous periods accumulate over time (i.e. being path-dependent).

While using empirical data to initialize, input, calibrate and validate the global trade model, we notice that the data availability varies greatly across countries, and it is the dataset available to all countries (the common denominator) that determines the data the model can use for input and validation. While some countries (e.g. UK, US) have more detailed or accurate data for trade and nutritional intake, the same data is not available for all countries, especially some countries in Africa, which largely restricts the data available for the global ABM. In some models, the problem is mitigated by grouping countries into regions and sub-regions, so missing data for some countries do not necessarily cause issues in the aggregated models. In ABM, however, because individual countries are modelled on their own, there is a higher demand for the same type of data with the same content, quality and format to be available for all countries. We identify the issue of data inequality across countries as one of the challenges in building an empirically grounded global ABM.

Finally, we find that a Pareto front can be a useful visual tool to show and compare the performance of models when the evaluation criteria are multi-dimensional. Not only can it identify a set of models with the 'best' performance (partially ordered), it can also reveal patterns of model performance in the evaluation space, especially when the number of candidate models is large. In this study, we select a model on the Pareto front by giving equal weights to the two evaluation criteria and pick the one with the highest weighted average. In the future, more sophisticated methods can be developed to select a model in the Pareto set; or one can include all models in the Pareto set and develop predictions based on multiple Pareto models, based on which new approaches can be developed towards model optimization, selection and prediction, which we leave for future research.

# 4. Preliminary results

## 4.1. The impact of trade on macro and micronutrient sufficiency in 2015

In this section, we show preliminary results on the impact of trade (saturation) on micro and macronutrient sufficiency. For each nutrient, we will compare the nutrition security of countries when trade saturation is low (60%, i.e. 60% of all potential trade opportunities are realized), medium (80%, i.e. 80% of all potential trade opportunities are realized) and high (100%, i.e. all potential trade opportunities are realized). Trade saturation is an exogenous intervention parameter that we will vary in the experiment. In reality, trade saturation can be influenced by countries' trade policies, trade conflicts and other factors such as a global pandemic. We will vary the level of trade saturation by low, medium and high in the computer experiment and study its impact on trade and food security.

As we discussed before in §2.2.1, the population-level nutrient intake (average per person) is calculated based on food consumption after production and trade, which we compare with nutrient requirement (average per person), which is based on the demographic compositions (age and sex) and varies across countries. If a country's average nutrient intake per person is higher than its average nutrient requirement per person, the country is considered secure in the specific nutrient, and vice versa.

For all graphs in this section except for 'fat', countries that are nutrient secure are in blue, and countries that are nutrient insecure are in red. Grading with the colours illustrates how far the country is from achieving nutrition security. The darker the shade of blue, the more secure the country is in that nutrient (intake ≫ maximum recommendation) and the darker the shade of red, the more insecure the country is in that nutrient (intake ≪ lower recommendation). For the nutrient 'fat', because the recommendation is within a range, there is a lower and upper limit (too little and too much fat can be can have negative health consequences), we use three colour schemes to show fat: red for below the lower recommendation, green for healthy, and yellow for over-consumption. In this section, we show the nutrient sufficiency in 2015 under low, medium and high trade saturations for calories, fat, vitamin A, iron and zinc, the latter three are important micronutrients, and in the Appendix, we will show that for folate, niacin, riboflavin, thiamine and vitamin C (figures 8–12).

Figure 3 shows the impact of trade on the consumption of calories. We see that with low to medium trade saturation, a handful of countries in Africa, Asia and South America will suffer from the lack of calories (or energy). The problem, however, can be solved by increasing trade saturation to 100%,

calories

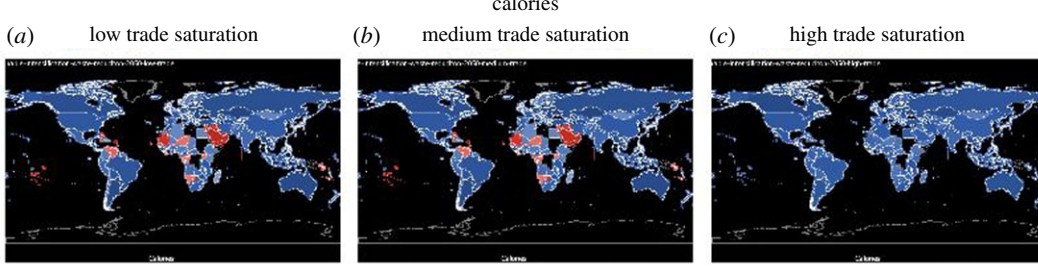

**Figure 3.** Calories sufficiency under low, medium and high trade saturation (blue = sufficient, red = insufficient).

fat

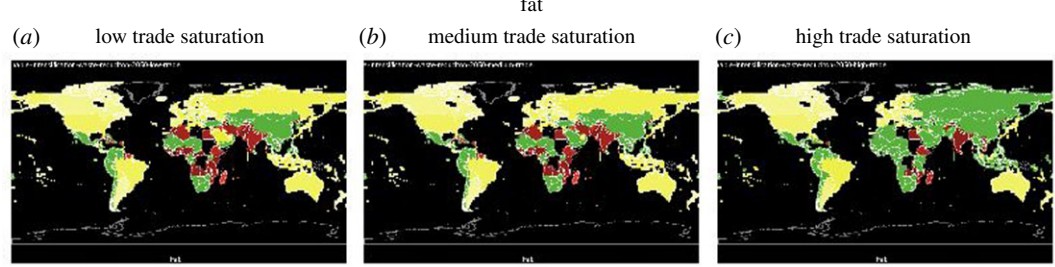

**Figure 4.** Fat sufficiency and over-consumption under low, medium and high trade saturation (green = healthy, red = insufficient, yellow = over-consumption).

vitamin A

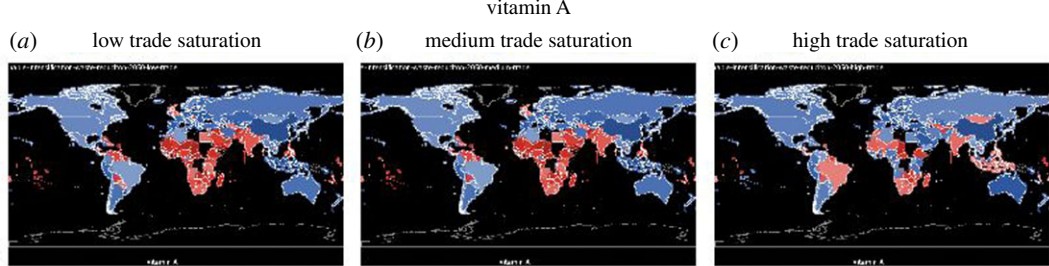

**Figure 5.** Vitamin A sufficiency under low, medium and high trade saturation (blue = sufficient, red = insufficient).

meaning that the world can produce food to feed the population with enough calories, as long as all trade opportunities are realized.

Figure 4 shows the impact of trade on the consumption of fat. We see that sufficient trade not only increases fat intake to within the healthy range in some countries in Africa, Asia and South America, and it also brings the fat intake down for some countries (e.g. Russia, Southeast Asian countries) and makes them healthier. The reasons that countries reduce fat intake when the trade is fully realized is that trade allows them to access a more diverse diet from a global supply.

Figure 5 shows the impact of trade saturation on vitamin A supply. We see more countries including developed ones such as the UK becoming deficient in vitamin A, an essential micronutrient, under low to medium trade saturation. Increasing trade saturation reduces the level of vitamin A deficiency to a large extent, especially in African countries. It does not, however, eliminate vitamin A deficiency altogether. Moreover, some countries that are not vitamin A deficient under low and medium trade saturation such as Brazil and Indonesia become so under full trade saturation, because some vitamin A-rich foods consumed domestically under low and medium trade can now be exported when trade saturation increases. Hence for vitamin A, although trade reduces the extent of deficiency in many countries, it alone cannot eliminate deficiency.

Figure 6 shows the impact of trade saturation on iron deficiency. As with vitamin A, more countries become deficient for iron than for calories, including some Scandinavian countries under the different levels of trade. The result shows that increasing trade saturation significantly reduces the level of iron deficiency, especially in countries in Africa, Asia and South America. However, trade does not eliminate iron deficiency in Scandinavia countries or Mongolia, although it makes them less deficient

iron

(*a*)    low trade saturation        (*b*)    medium trade saturation        (*c*)    high trade saturation

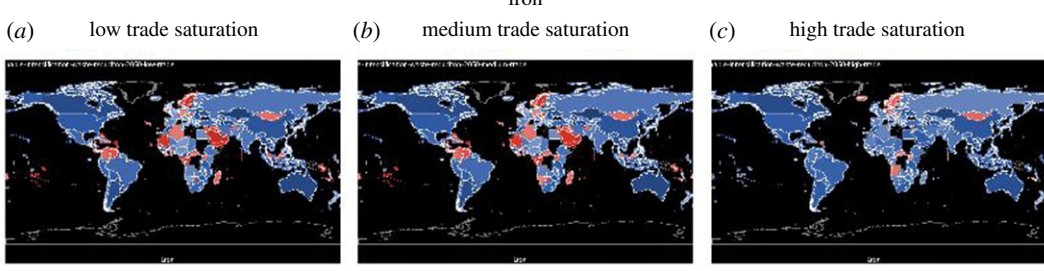

**Figure 6.** Iron sufficiency under low, medium and high trade saturation (blue = sufficient, red = insufficient).

zinc

(*a*)    low trade saturation        (*b*)    medium trade saturation        (*c*)    high trade saturation

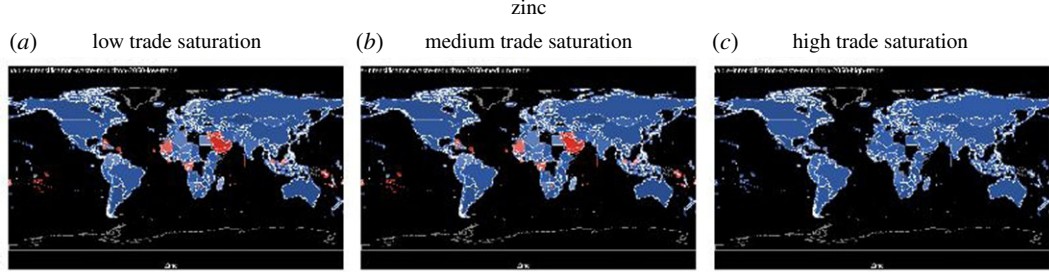

**Figure 7.** Zinc sufficiency under low, medium and high trade saturation (blue = sufficient, red = insufficient).

(light shade in red). As with vitamin A, other measures such as improving diet and increasing production and consumption of certain foods will be needed to eliminate iron deficiency.

Figure 7 shows the impact of trade saturation on zinc deficiency. It shows that increasing trade saturation from low or medium to high can eliminate zinc deficiency.

The results show that deficiency in micronutrients such as vitamin A and iron is much more common than that in macronutrients such as calories and fat. The latter is found only in developing countries in Africa, Asia and South America while the former can be found in both developed and developing countries, which signifies the importance of including both macro and micronutrients in the study of food and nutrition security.

We see that trade plays a positive role in improving food and nutrition security in almost all cases. A fully realized trade eliminates calories deficiency in all countries, compared with low trade and makes fat intakes more balanced across countries. For some micronutrients (e.g. niacin and zinc, see Appendix), high trade saturation has successfully eliminated deficiency in all countries, compared with low and medium trade saturation; for others, deficiency still exists even under fully realized trade, but to a lesser degree. It will require other measures, such as change increasing production and alternative trade scenarios of food rich in the micronutrients to eliminate nutrient deficiency in those micronutrients. In many developing countries currently use supplementation to correct the deficiencies. In almost all cases, we observe that trade at least reduces the level of nutrient deficiency across countries, if not eliminates it. The effect of trade is most prominent in countries most vulnerable to food and nutrition insecurity, including those in Africa, Asia and South America.

Finally, table 6 lists the nutrition intake per capita of the 20 most populous countries in 2013, which together account for more than two-thirds of the world population; also table 22 in the Appendix for the full list of countries. Note that the nutrient requirement per capita will vary slightly by countries due to the different population composition in each country (e.g. a younger/higher male percentage population will require more nutrient).

## 4.2. Discussion

The preliminary results from the model demonstrate the impact of increasing trade on improving macro and micronutrient sufficiency across countries in the world. For all nutrients included in §4.1, fully realized global trade can help ensure a sufficient supply of nutrients and enables countries to have a healthier diet. The reason is quite straightforward: trade allows the countries to access a larger variety of food, thus enabling them to have a more nutritionally balanced diet. Trade is even more important

**Table 6.** Nutrition intake per capita[a] of the 20 most populous countries in 2013, which together account for more than two-thirds of the world population (also table 22 in the Appendix for the full list of countries).

| country | pop, millions | calories | protein | fat | vitamin C | vitamin A | folate | iron | zinc | thiamine | riboflavin | niacin | saturated FA |
|---|---|---|---|---|---|---|---|---|---|---|---|---|---|
| China | 1384 | 2750 | 89.9 | 85.7 | 275.4 | 1686.9 | 575.9 | 26.3 | 12.3 | 2.60 | 2.06 | 23.77 | 25.32 |
| India | 1296 | 2543 | 62.4 | 55.3 | 91.4 | 568.3 | 392.8 | 17.0 | 8.3 | 1.63 | 1.27 | 14.23 | 19.73 |
| United States | 329 | 3869 | 105.1 | 188.6 | 103.1 | 741.6 | 752.1 | 20.0 | 14.3 | 2.64 | 3.38 | 32.18 | 63.68 |
| Indonesia | 262 | 2967 | 65.7 | 63.8 | 84.5 | 546.2 | 410.8 | 22.9 | 10.4 | 2.83 | 1.33 | 26.53 | 28.92 |
| Brazil | 208 | 3503 | 99.7 | 131.6 | 95.7 | 572.2 | 790.5 | 22.3 | 13.2 | 2.75 | 2.53 | 29.57 | 48.94 |
| Pakistan | 207 | 2738 | 72.4 | 82.5 | 39.6 | 794.9 | 258.8 | 10.4 | 8.1 | 0.94 | 1.66 | 10.24 | 31.98 |
| Nigeria | 203 | 3158 | 72.2 | 64.9 | 181.8 | 674.6 | 703.5 | 23.0 | 11.4 | 2.90 | 1.37 | 25.53 | 20.84 |
| Bangladesh | 159 | 2387 | 53.4 | 29.0 | 40.1 | 331.9 | 187.9 | 19.3 | 8.5 | 2.25 | 0.86 | 19.34 | 9.31 |
| Russia | 142 | 3347 | 93.7 | 117.0 | 102.5 | 810.5 | 314.4 | 13.5 | 10.7 | 1.39 | 2.04 | 18.99 | 36.70 |
| Japan | 126 | 2525 | 75.3 | 93.6 | 89.9 | 712.5 | 314.1 | 17.3 | 10.1 | 1.61 | 1.81 | 17.67 | 26.39 |
| Mexico | 125 | 3295 | 93.4 | 103.6 | 102.5 | 787.1 | 618.1 | 21.9 | 14.4 | 2.76 | 2.46 | 29.24 | 34.04 |
| Ethiopia | 108 | 2108 | 66.6 | 33.3 | 37.8 | 443.7 | 417.1 | 18.4 | 11.8 | 1.92 | 1.15 | 18.13 | 9.48 |
| Philippines | 105 | 2710 | 63.2 | 56.0 | 108.5 | 486.7 | 375.3 | 19.9 | 9.1 | 2.63 | 1.42 | 25.07 | 24.68 |
| Egypt | 99 | 3247 | 91.2 | 57.2 | 127.7 | 712.6 | 386.6 | 20.4 | 11.4 | 1.86 | 1.70 | 20.99 | 16.86 |
| Vietnam | 97 | 2823 | 80.1 | 71.3 | 155.0 | 885.4 | 451.3 | 27.4 | 11.7 | 3.11 | 1.68 | 27.14 | 24.26 |
| Congo | 85 | 2306 | 48.2 | 46.8 | 163.7 | 583.2 | 583.1 | 12.9 | 6.4 | 2.08 | 0.95 | 20.89 | 14.38 |
| Iran | 83 | 3478 | 98.9 | 82.0 | 217.3 | 997.7 | 553.7 | 20.9 | 10.8 | 1.83 | 1.87 | 19.67 | 21.87 |
| Turkey | 81 | 3956 | 111.6 | 129.1 | 174.2 | 1177.6 | 582.0 | 20.5 | 13.1 | 1.75 | 2.65 | 19.23 | 38.42 |
| Germany | 80 | 3787 | 94.8 | 180.8 | 100.3 | 998.6 | 298.8 | 12.8 | 12.0 | 1.65 | 2.87 | 19.32 | 63.85 |
| Thailand | 68 | 2786 | 60.6 | 57.7 | 120.2 | 624.9 | 244.6 | 21.5 | 8.8 | 1.95 | 1.20 | 19.57 | 21.43 |

[a]nutrition intake per capita will be compared with the nutrition required per capita in the country, which depends on the country's demographic composition to determine a country's food and nutrition security.

when considering micronutrient sufficiency because they tend to be more concentrated in specific foods than macronutrients such as calories and fat, so the diet needs to contain more diverse food items to supply sufficient micronutrients. The role of increasing trade in improving nutrition security is found to be universal across all macro and micronutrients, but more pronounced in micronutrients. Results from this study agree with previous studies in that global trade can balance food supply and demand across regions and smooth out nutrient intake across countries [60]. It also confirms the previous conclusion that the focus of food security should shift from calories to critical micronutrients, in which many regions will continue to have inadequacies [41].

A criticism of global food trade is that free trade penalizes the poorest and most vulnerable countries [61,62]. This is largely true: the results show that low-income countries are the first to fall into food and nutrition insecurity as trade saturation goes from high to medium to low, partly because they are often ranked lower on countries' trade priority due to their low GDP per capita and remote location. However, the results also suggest that the solution is not trade-protectionism. This is very important for countries where diets need to diversity to achieve nutrition security. We need to further remove any trade barriers and increase trade saturation so that all potential trade opportunities can be used and the demand for the most vulnerable countries can be met. Preliminary results show that increasing trade saturation always improves the nutrition security of countries. For some nutrients, trade alone can eliminate food and nutrition insecurity for all countries altogether. The poor and vulnerable countries also benefit from trade. Because they are lower on the priority rank, they benefit the most from additional trade liberalization and opportunities. Pradhan *et al*. [63] estimate that in 2000 about 1 billion people from Asia and Africa require cross-continental agricultural trade to be food secure, and by 2050 the number of people depending on trade for food security will be between 1.5 and 6 billion. Any hindrance to trade will put these people at high risk of food insecurity.

The model presented in the study is the first step towards a holistic approach to the grand challenge of food and nutrition security under global trade and climate change. There are many ways to extend the model in the future. First, the dimension and factors of countries' trade relationships included in the model are far from complete. Apart from GDP, distance, and historic and emergent trade relationships, many factors can affect the trade relationships between countries. For example, the trade war between the US and China in 2019 was motivated by complex economic and political reasons. The flexibility of the ABM, however, means that the model can be easily adapted to implement additional factors to increase the dimension of the trade relationship. The relation-driven trade model developed in this paper will be a framework to which these relation-driven dimensions and motivations can be added to analyse its impact on trade and food security.

Moreover, so far we have only considered the average nutrition intake in a country. We have not considered the inequality and distribution within a country. A country may have enough food for its population overall, but is still food insecure because of unequal internal allocation. This is especially important for countries with unequal developments across regions, where a minority of the population has more than adequate access to food and nutrition while the majority do not. Since the global model cannot model each country in such details due to computational constraints, one approach is to use a measure of inequality of access to food within a country as used by the FAO, such as in the FEEDME model [64,65], which uses a country-level measure of inequality (income and food access inequality) to determine the percentage of a population in each country which is undernourished. Another approach is to couple the global model with individual country models for the country of interest. For example, we can couple the individual country model of China or Brazil with the global trade model developed in this paper, which allows us to position the internal dynamics and interactions between different regions and firms under the background of global trade, without having to acquire the same level of detailed data for all countries. This is particularly useful for countries like Brazil where global food trade plays a crucial role in its domestic agri-food systems.

# 5. Concluding remarks

In an increasingly connected world, a country's ability to secure food concerns more than what is produced on the ground, but also its relationships with other countries. The issue is particularly relevant and timely given the current debates about Brexit, US–China trade war and the global COVID-19 pandemic. This paper develops an empirically grounded, relation-driven model of global trade to study the impact of trade on the food and nutrition security of countries around the world, complementing existing price-driven trade models in the literature. It makes several important

contributions. First, it provides a flexible modelling framework that focuses on country-wise relationships in trade. The framework can be used to study more complex and relation-driven behaviour when trading with each other, such as trust, familiarity, history and conflicts. Second, the global trade model is linked with a comprehensive nutrition formula based on food consumption to investigate the impact of trade on food and nutrition security. As researchers have been constantly calling for a holistic and multifactorial approach to these global challenges, the model is one of the first attempts to develop an integrated framework consisting of socio-economic, geopolitical, nutrition, environmental and agri-food systems to tackle these global challenges.

Preliminary results show that global trade has a significant impact on food and nutrition security across countries. Increasing trade improves the nutritional security of almost all countries, especially countries in Africa, Asia and Latin America susceptible to food insecurity. For some nutrients such as calories, niacin and zinc, trade alone can eliminate nutrition insecurity for all countries; for others such as vitamin A, folate, riboflavin and iron, trade improves nutrition security but is insufficient to achieve food security. Other measures such as dietary changes may be needed. We also find that trade allows countries to have a healthier and more balanced diet, due to the increased variety of food enabled by trade: it decreases fat intake in countries that previously consume too much fat and vice versa. Overall, we find that the effect of trade on enhancing nutrition balance and security is universal across all macro and micronutrients and countries.

In the future, the global trade model developed in the study can be coupled with individual country models to include more detailed regional dynamics, firm interactions and inequality within countries. It offers a way to link models of individual country's agri-food system with a dynamic global trade system, which is particularly useful for countries like Brazil where global food trade plays a crucial role. Finally, the model will be used to systemically conduct scenario analysis on food security under various scenarios of climate change, dietary change and de-globalization.

Data accessibility. We have provided the (simulated) data used in the paper as electronic supplementary material.

Authors' contributions. J.G. was involved in conceptualization, formal analysis, investigation, methodology, validation, visualization and writing the original draft preparation. J.G.P. was involved in conceptualization, methodology, writing review and editing. J.I.M. was involved in conceptualization, data curation, methodology, funding acquisition, writing' review and editing. N.F. was involved in data curation, formal analysis, writing review and editing. P.S. was involved in conceptualization, data curation, funding acquisition, methodology, writing review and editing. H.C. was involved in data curation, formal analysis, writing review and editing. T.D. was involved in conceptualization, funding acquisition, writing review and editing. M.A. was involved in conceptualization.

Competing interests. P.S. was a member of the Royal Society Editorial Board at the time this paper was submitted, peer-reviewed and accepted; however, P.S. had no involvement in the assessment of this manuscript.

Funding. The Scottish Government's Environment, Agriculture and Food Strategic Research Portfolio and the Belmont Forum/FACCE-JPI (NERC grant no. NE/M021327/1) funded this research.

Acknowledgements. We would like to thank two anonymous reviewers for their constructive comments and suggestions, which help us greatly improve the paper.

# Appendix A

## A.1. Validation

The data we use for validation are the same as for calibration, except that we use the year 2001–2007 for calibration and 2008–2013 for validation. We run the model 30 times with the calibrated parameters in table 5 and calculate the mean and standard deviation (s.d.) of the match rate in trade volume and trade partners among the 30 runs. The full validation results are available in tables 15 to 21.

### A.1.1. Trade volume

Table 7 shows the percentage of the simulated volume of trade and consumption that is within ±20% of the actual volume from data. We see that, overall, the model produces a 54.80% matching rate. It does better in predicting export volume and food consumption than predicting import volume. The variance or s.d. among model runs (from internal model stochasticity) is very small, which suggests the absence of bifurcation.

Table 8 shows the matching rate of simulated volume by aggregated food category. Note that the average match rate varies across different food categories. The model's prediction of trade and consumption volumes is better for some foods (sugar crops, spices, animal fats, aquatic products) than

**Table 7.** Overall match for trade *volume* (within ±20% of actual volume).

| | mean | s.d. |
|---|---|---|
| all | 0.5480 | 0.0009 |
| import | 0.4757 | 0.0011 |
| export | 0.6177 | 0.0009 |
| food | 0.5507 | 0.0008 |

**Table 8.** Match rate for trade *volume* by commodity.

| | mean | s.d. |
|---|---|---|
| cereals—excluding beer | 0.4915 | 0.0014 |
| starchy roots | 0.6046 | 0.0002 |
| sugar crops | 0.7997 | 0.0001 |
| sugar & sweeteners | 0.5569 | 0.0012 |
| pulses | 0.5024 | 0.0015 |
| treenuts | 0.6247 | 0.0009 |
| oilcrops | 0.5790 | 0.0014 |
| vegetable oils | 0.6054 | 0.0010 |
| vegetables | 0.3512 | 0.0006 |
| fruits—excluding wine | 0.4777 | 0.0006 |
| stimulants | 0.4134 | 0.0017 |
| spices | 0.6942 | 0.0009 |
| alcoholic beverages | 0.4336 | 0.0011 |
| meat | 0.4492 | 0.0015 |
| offals | 0.4978 | 0.0016 |
| animal fats | 0.6663 | 0.0008 |
| milk—excluding butter | 0.3212 | 0.0011 |
| eggs | 0.4601 | 0.0019 |
| fish; seafood | 0.4580 | 0.0008 |
| aquatic products; other | 0.8777 | 0.0004 |

others (vegetables, milk). Also note that the variance or s.d. tends to be higher for foods with lower average matching rate, which suggests that those foods are more subject to stochastic factors in trade, which may explain the lower matching rate in prediction.

Table 9 shows the matching rate of simulated volume by region. The model can better predict trade volumes for some regions (sub-Saharan Africa, Latin America) than others (North America, Europe, Pacific OECD). The regions that the model does relatively well in predicting tend to be developing regions and countries (sub-Saharan Africa, Latin America); the regions that the model does relatively poorly are developed ones (North America, Europe and Pacific OECD). The model's assumption of increasing consumption as production increases may be more valid in developing countries than developed ones; it may not capture the most recent trend in food consumption in developed countries that diverge from a concave increasing function as projected by most conventional models. Empirical data and evidence would be needed to adjust the model for that.

Table 10 shows the matching rate of simulated volume by year. We can see that the matching rate steadily goes down over the years, which is as expected as we move away from the original year (2000) initialized with empirical data. The variance of results from the parallel models also increases over time, which is also as expected, because stochasticity (such as emergent trade relationship)

**Table 9.** Match rate for trade *volume* by region.

|  | mean | s.d. |
| --- | --- | --- |
| sub-Saharan Africa | 0.6253 | 0.0001 |
| centrally planned Asia | 0.5473 | 0.0004 |
| Europe | 0.4338 | 0.0001 |
| former Soviet Union | 0.5285 | 0.0003 |
| Latin America | 0.5955 | 0.0002 |
| Middle East/North Africa | 0.4674 | 0.0001 |
| North America | 0.3260 | 0.0004 |
| Pacific OECD | 0.4351 | 0.0003 |
| Pacific Asia | 0.5563 | 0.0003 |
| South Asia | 0.5349 | 0.0002 |

**Table 10.** Match rate for trade *volume* by year.

|  | mean | s.d. |
| --- | --- | --- |
| year2008 | 0.5689 | 0.0009 |
| year2009 | 0.5642 | 0.0009 |
| year2010 | 0.5526 | 0.0009 |
| year2011 | 0.5419 | 0.0009 |
| year2012 | 0.5326 | 0.0010 |
| year2013 | 0.5279 | 0.0011 |

occuring in early years may have a lasting effect on model results from that point, the accumulation of which will lead to increased variance among parallel models.

## A.1.2. Trade partners

This section shows the matching rate of trade partners between simulated and actual trade outcome (importer, exporter and food traded). Table 11 shows the overall matching rate of trade partners and matching rate by year. Overall, the model correctly predicts 67.31% of all trade partners between 2008 and 2013. Similar to trade volume, the model's predictive power decreases over time from 68.95% in 2008 to 66.90% in 2013. The decrease, however, is moderate and the model still has a relatively good predictive power in 2013.

**Table 11.** Match rate for trade *partner* by year.

|  | mean | s.d. |
| --- | --- | --- |
| overall | 0.6731 | 0.0005 |
| year2008 | 0.6895 | 0.0008 |
| year2009 | 0.6815 | 0.0009 |
| year2010 | 0.6584 | 0.0010 |
| year2011 | 0.6703 | 0.0011 |
| year2012 | 0.6697 | 0.0011 |
| year2013 | 0.6689 | 0.0008 |

**Table 12.** Match rate for trade *partner* by commodity.

|  | mean | s.d. |
|---|---|---|
| cereals—excluding beer | 0.6891 | 0.0006 |
| starchy roots | 0.3879 | 0.0020 |
| sugar crops | 0.7139 | 0.0006 |
| pulses | 0.5391 | 0.0012 |
| treenuts | 0.6651 | 0.0017 |
| oilcrops | 0.9020 | 0.0004 |
| vegetables | 0.7121 | 0.0038 |
| fruits—excluding wine | 0.6770 | 0.0011 |
| stimulants | 0.9167 | 0.0023 |
| spices | 0.8770 | 0.0086 |

Table 12 shows the matching rate of trade partners by aggregated food category. Because the data used for validation (from Comtrade) does not cover all the food traded in the model, only the ones available in Comtrade are presented here. Similar to trade volume, the model's ability to predict trade partners varies across food categories. For some (e.g. oil crops, stimulants and spices), the model can predict correctly the majority (greater than 85%) of the trade partners, and for others (starchy roots), the model can only predict less than half. One of the reasons could be that the production of some commodities is concentrated in a handful of countries (e.g. oil crops, stimulants and spices). It is relatively easy for the model to predict correctly trade partners for those commodities, as there are only a handful of sellers, than for commodities where buyers and sellers are more scattered (tables 13–22).

vitamin C

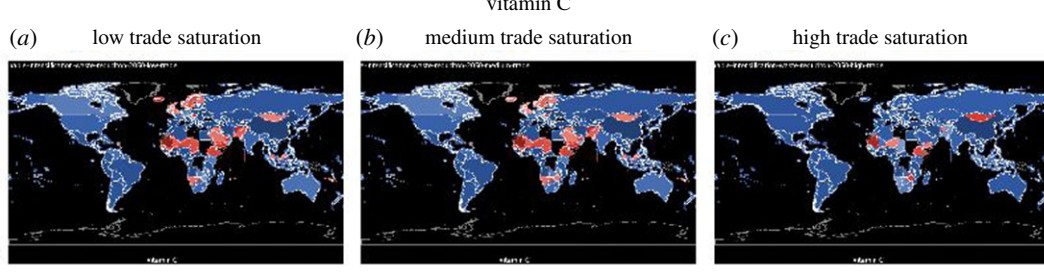

**Figure 8.** Vitamin C sufficiency under low, medium and high trade saturation (blue = sufficient, red = insufficient).

folate

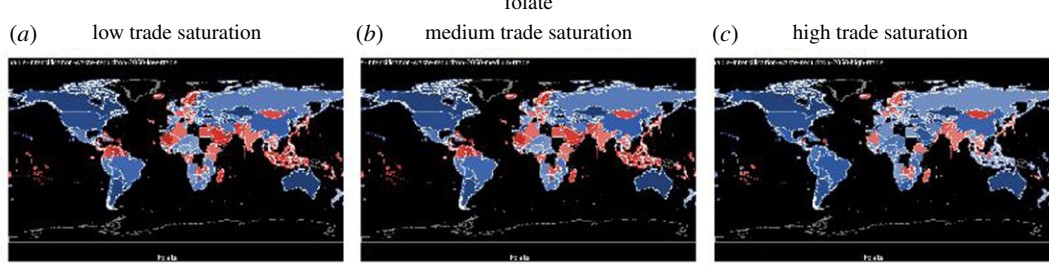

**Figure 9.** Folate sufficiency under low, medium and high trade saturation (blue = sufficient, red = insufficient).

niacin

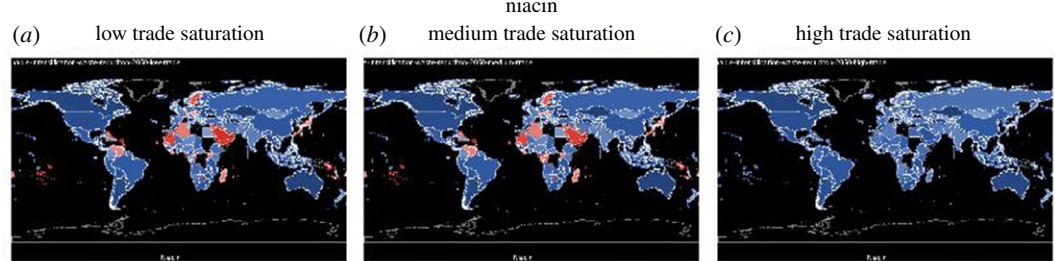

**Figure 10.** Niacin sufficiency under low, medium and high trade saturation (blue = sufficient, red = insufficient).

riboflavin

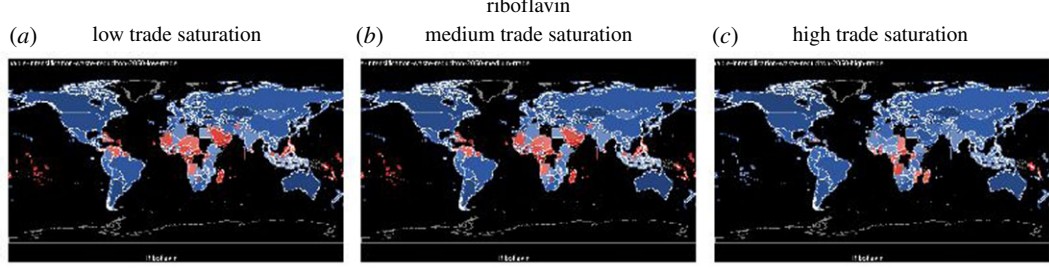

**Figure 11.** Riboflavin sufficiency under low, medium and high trade saturation (blue = sufficient, red = insufficient).

thiamin

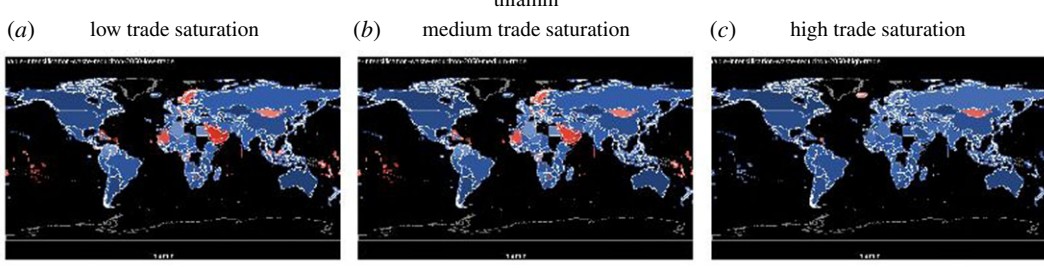

**Figure 12** Thiamine sufficiency under low, medium and high trade saturation (blue = sufficient, red = insufficient).

**Table 13.** Example data of simulated versus actual trade *volumes* (see electronic supplementary material for full results).

| country | element | item | year | actual | simulated |
|---|---|---|---|---|---|
| Afghanistan | import | wheat and products | 2000 | 929 | 1570 |
| Afghanistan | export | wheat and products | 2000 | 0 | 0 |
| Afghanistan | food | wheat and products | 2000 | 2600 | 2681 |
| Afghanistan | import | rice (milled equivalent) | 2000 | 219 | 260 |
| Afghanistan | export | rice (milled equivalent) | 2000 | 0 | 0 |
| Afghanistan | food | rice (milled equivalent) | 2000 | 372 | 410 |
| Afghanistan | import | barley and products | 2000 | 5 | 90 |
| Afghanistan | export | barley and products | 2000 | 0 | 0 |
| Afghanistan | food | barley and products | 2000 | 84 | 95 |
| Afghanistan | import | maize and products | 2000 | 0 | 63 |
| Afghanistan | export | maize and products | 2000 | 0 | 0 |
| Afghanistan | food | maize and products | 2000 | 66 | 109 |
| Afghanistan | import | rye and products | 2000 | 0 | 0 |
| Afghanistan | export | rye and products | 2000 | 0 | 0 |
| Afghanistan | food | rye and products | 2000 | 0 | 0 |
| Afghanistan | import | oats | 2000 | 0 | 0 |
| Afghanistan | export | oats | 2000 | 0 | 0 |
| Afghanistan | food | oats | 2000 | 0 | 0 |
| Afghanistan | import | millet and products | 2000 | 0 | 0 |
| Afghanistan | export | millet and products | 2000 | 0 | 0.67 |
| Afghanistan | food | millet and products | 2000 | 20 | 19 |
| Afghanistan | import | sorghum and products | 2000 | 0 | 0 |
| Afghanistan | export | sorghum and products | 2000 | 0 | 0 |
| Afghanistan | food | sorghum and products | 2000 | 0 | 0 |
| Afghanistan | import | cereals; other | 2000 | 0 | 0 |
| Afghanistan | export | cereals; other | 2000 | 0 | 0 |
| Afghanistan | food | cereals; other | 2000 | 0 | 0 |
| Afghanistan | import | cassava and products | 2000 | 0 | 0 |
| Afghanistan | export | cassava and products | 2000 | 0 | 0 |
| Afghanistan | food | cassava and products | 2000 | 0 | 0 |

**Table 14.** Example data of simulated versus actual trade *partners* (see electronic supplementary material for full results).

| item | year | from | to |
| --- | --- | --- | --- |
| wheat and products | 2000 | India | Malaysia |
| wheat and products | 2000 | India | Sri Lanka |
| wheat and products | 2000 | India | Nepal |
| wheat and products | 2000 | India | Mexico |
| wheat and products | 2000 | India | Russian Federation |
| wheat and products | 2000 | India | Ukraine |
| wheat and products | 2000 | India | Iraq |
| wheat and products | 2000 | India | Italy |
| wheat and products | 2000 | India | Ethiopia |
| wheat and products | 2000 | India | Netherlands |
| wheat and products | 2000 | India | Philippines |
| wheat and products | 2000 | India | Bangladesh |
| wheat and products | 2000 | India | Brazil |
| wheat and products | 2000 | India | Algeria |
| wheat and products | 2000 | India | Republic of Korea |
| wheat and products | 2000 | India | Nigeria |
| wheat and products | 2000 | India | China |
| wheat and products | 2000 | India | Thailand |
| wheat and products | 2000 | India | Afghanistan |
| wheat and products | 2000 | India | United Arab Emirates |
| wheat and products | 2000 | India | Israel |
| wheat and products | 2000 | India | Japan |
| wheat and products | 2000 | India | Morocco |
| wheat and products | 2000 | India | Yemen |
| wheat and products | 2000 | India | Indonesia |
| wheat and products | 2000 | India | Iran (Islamic Republic of) |
| wheat and products | 2000 | India | Egypt |
| wheat and products | 2000 | Denmark | Belgium |
| wheat and products | 2000 | Denmark | Italy |
| wheat and products | 2000 | Denmark | Netherlands |
| wheat and products | 2000 | Denmark | Republic of Korea |
| wheat and products | 2000 | Denmark | Japan |
| wheat and products | 2000 | Denmark | Luxembourg |
| wheat and products | 2000 | Denmark | Israel |
| wheat and products | 2000 | Canada | Viet Nam |
| wheat and products | 2000 | Canada | Colombia |
| wheat and products | 2000 | Canada | Chile |
| wheat and products | 2000 | Canada | Malaysia |
| wheat and products | 2000 | Canada | Jamaica |
| wheat and products | 2000 | Canada | Indonesia |
| wheat and products | 2000 | Canada | El Salvador |
| wheat and products | 2000 | Canada | Panama |

**Table 15.** The matching rate of the calibrated model for all trade *volumes*, and *volumes* for import, export and food consumption.

| index-run | all | import | export | food |
|---|---|---|---|---|
| 01 | 0.548832 | 0.476725 | 0.618373 | 0.551398 |
| 02 | 0.547023 | 0.474583 | 0.616645 | 0.54984 |
| 03 | 0.547062 | 0.474551 | 0.616891 | 0.549743 |
| 04 | 0.548839 | 0.476715 | 0.618437 | 0.551366 |
| 05 | 0.54699 | 0.474551 | 0.616667 | 0.549754 |
| 06 | 0.546823 | 0.474422 | 0.616389 | 0.549658 |
| 07 | 0.546958 | 0.47454 | 0.616592 | 0.549743 |
| 08 | 0.548797 | 0.476704 | 0.618363 | 0.551324 |
| 09 | 0.547097 | 0.474604 | 0.616859 | 0.549829 |
| 10 | 0.547087 | 0.474529 | 0.616913 | 0.549818 |
| 11 | 0.54688 | 0.474487 | 0.616453 | 0.5497 |
| 12 | 0.548885 | 0.476704 | 0.61848 | 0.551473 |
| 13 | 0.548793 | 0.476651 | 0.618426 | 0.551302 |
| 14 | 0.548924 | 0.476672 | 0.618671 | 0.55143 |
| 15 | 0.548839 | 0.476683 | 0.618384 | 0.551451 |
| 16 | 0.54885 | 0.476736 | 0.618352 | 0.551462 |
| 17 | 0.54699 | 0.474551 | 0.616677 | 0.549743 |
| 18 | 0.54891 | 0.476683 | 0.618596 | 0.551451 |
| 19 | 0.546994 | 0.474476 | 0.616752 | 0.549754 |
| 20 | 0.548903 | 0.476704 | 0.618565 | 0.551441 |
| 21 | 0.548761 | 0.476619 | 0.618416 | 0.551249 |
| 22 | 0.548761 | 0.476629 | 0.618278 | 0.551377 |
| 23 | 0.548871 | 0.476608 | 0.618628 | 0.551377 |
| 24 | 0.548868 | 0.476683 | 0.618501 | 0.551419 |
| 25 | 0.54703 | 0.474572 | 0.616699 | 0.549818 |
| 26 | 0.546933 | 0.474433 | 0.616667 | 0.5497 |
| 27 | 0.548956 | 0.476746 | 0.618639 | 0.551483 |
| 28 | 0.548786 | 0.476651 | 0.618394 | 0.551313 |
| 29 | 0.548751 | 0.476704 | 0.618235 | 0.551313 |
| 30 | 0.546891 | 0.474454 | 0.616602 | 0.549615 |
| mean | 0.548036 | 0.475746 | 0.617685 | 0.550678 |
| s.d. | 0.000941 | 0.001092 | 0.000907 | 0.000831 |

**Table 16.** The matching rate of the calibrated model for trade *volumes* for different foods (part 1).

| run | cereals | starchy roots | sugar crops | sugar & sweeteners | pulses | treenuts | oilcrops | vegetable oils | vegetables | fruits |
|---|---|---|---|---|---|---|---|---|---|---|
| 01 | 0.4927 | 0.6046 | 0.7998 | 0.5582 | 0.5035 | 0.6255 | 0.5798 | 0.6061 | 0.3512 | 0.4783 |
| 02 | 0.4900 | 0.6047 | 0.7996 | 0.5552 | 0.5003 | 0.6239 | 0.5772 | 0.6045 | 0.3509 | 0.4770 |
| 03 | 0.4899 | 0.6047 | 0.7996 | 0.5560 | 0.5008 | 0.6235 | 0.5779 | 0.6041 | 0.3508 | 0.4770 |
| 04 | 0.4927 | 0.6047 | 0.7998 | 0.5577 | 0.5037 | 0.6253 | 0.5800 | 0.6060 | 0.3515 | 0.4783 |
| 05 | 0.4899 | 0.6043 | 0.7996 | 0.5555 | 0.5008 | 0.6235 | 0.5779 | 0.6044 | 0.3511 | 0.4770 |
| 06 | 0.4900 | 0.6041 | 0.7996 | 0.5550 | 0.5006 | 0.6236 | 0.5776 | 0.6039 | 0.3502 | 0.4771 |
| 07 | 0.4900 | 0.6043 | 0.7996 | 0.5558 | 0.5005 | 0.6237 | 0.5776 | 0.6044 | 0.3499 | 0.4770 |
| 08 | 0.4927 | 0.6046 | 0.7998 | 0.5582 | 0.5037 | 0.6253 | 0.5800 | 0.6061 | 0.3516 | 0.4781 |
| 09 | 0.4899 | 0.6045 | 0.7996 | 0.5555 | 0.5012 | 0.6237 | 0.5769 | 0.6048 | 0.3508 | 0.4771 |
| 10 | 0.4898 | 0.6047 | 0.7996 | 0.5560 | 0.5008 | 0.6238 | 0.5772 | 0.6047 | 0.3514 | 0.4768 |
| 11 | 0.4896 | 0.6044 | 0.7996 | 0.5558 | 0.5006 | 0.6236 | 0.5776 | 0.6039 | 0.3500 | 0.4769 |
| 12 | 0.4928 | 0.6046 | 0.7998 | 0.5578 | 0.5033 | 0.6254 | 0.5801 | 0.6066 | 0.3510 | 0.4781 |
| 13 | 0.4926 | 0.6048 | 0.7998 | 0.5577 | 0.5035 | 0.6254 | 0.5800 | 0.6060 | 0.3514 | 0.4781 |
| 14 | 0.4928 | 0.6046 | 0.7998 | 0.5581 | 0.5040 | 0.6255 | 0.5803 | 0.6061 | 0.3517 | 0.4780 |
| 15 | 0.4928 | 0.6046 | 0.7998 | 0.5575 | 0.5036 | 0.6254 | 0.5803 | 0.6063 | 0.3515 | 0.4782 |
| 16 | 0.4927 | 0.6048 | 0.7998 | 0.5578 | 0.5038 | 0.6254 | 0.5805 | 0.6063 | 0.3514 | 0.4782 |
| 17 | 0.4900 | 0.6042 | 0.7996 | 0.5558 | 0.5006 | 0.6235 | 0.5776 | 0.6042 | 0.3511 | 0.4770 |
| 18 | 0.4927 | 0.6042 | 0.7998 | 0.5581 | 0.5041 | 0.6253 | 0.5803 | 0.6064 | 0.3516 | 0.4782 |
| 19 | 0.4898 | 0.6042 | 0.7996 | 0.5555 | 0.5007 | 0.6235 | 0.5776 | 0.6044 | 0.3516 | 0.4771 |
| 20 | 0.4926 | 0.6047 | 0.7998 | 0.5577 | 0.5035 | 0.6255 | 0.5801 | 0.6066 | 0.3510 | 0.4784 |
| 21 | 0.4927 | 0.6045 | 0.7998 | 0.5578 | 0.5039 | 0.6254 | 0.5801 | 0.6057 | 0.3516 | 0.4782 |
| 22 | 0.4928 | 0.6048 | 0.7998 | 0.5577 | 0.5036 | 0.6255 | 0.5800 | 0.6060 | 0.3517 | 0.4780 |
| 23 | 0.4929 | 0.6048 | 0.7998 | 0.5579 | 0.5041 | 0.6255 | 0.5801 | 0.6059 | 0.3517 | 0.4781 |

(Continued.)

**Table 16.** (*Continued.*)

| run | cereals | starchy roots | sugar crops | sugar & sweeteners | pulses | treenuts | oilcrops | vegetable oils | vegetables | fruits |
| --- | --- | --- | --- | --- | --- | --- | --- | --- | --- | --- |
| 24 | 0.4927 | 0.6048 | 0.7998 | 0.5572 | 0.5040 | 0.6256 | 0.5803 | 0.6062 | 0.3520 | 0.4782 |
| 25 | 0.4898 | 0.6043 | 0.7996 | 0.5557 | 0.5006 | 0.6237 | 0.5772 | 0.6045 | 0.3513 | 0.4769 |
| 26 | 0.4900 | 0.6045 | 0.7996 | 0.5553 | 0.5008 | 0.6236 | 0.5777 | 0.6041 | 0.3507 | 0.4771 |
| 27 | 0.4927 | 0.6046 | 0.7998 | 0.5583 | 0.5037 | 0.6254 | 0.5800 | 0.6065 | 0.3529 | 0.4784 |
| 28 | 0.4926 | 0.6045 | 0.7998 | 0.5578 | 0.5039 | 0.6254 | 0.5801 | 0.6060 | 0.3502 | 0.4784 |
| 29 | 0.4928 | 0.6048 | 0.7998 | 0.5580 | 0.5033 | 0.6255 | 0.5798 | 0.6060 | 0.3510 | 0.4782 |
| 30 | 0.4898 | 0.6045 | 0.7996 | 0.5553 | 0.5010 | 0.6239 | 0.5771 | 0.6040 | 0.3505 | 0.4769 |
| mean | 0.4915 | 0.6046 | 0.7997 | 0.5569 | 0.5024 | 0.6247 | 0.5790 | 0.6054 | 0.3512 | 0.4777 |
| s.d. | 0.0014 | 0.0002 | 0.0001 | 0.0012 | 0.0015 | 0.0009 | 0.0014 | 0.0010 | 0.0006 | 0.0006 |

**Table 17.** The matching rate of the calibrated model for trade *volumes* for different foods (part 2).

| run | stimulants | spices | alcoholic beverages | meat | offals | animal fats | milk—excluding butter | eggs | fish; seafood | aquatic products |
|---|---|---|---|---|---|---|---|---|---|---|
| 01 | 0.4150 | 0.6949 | 0.4346 | 0.4506 | 0.4987 | 0.6672 | 0.3222 | 0.4596 | 0.4587 | 0.8778 |
| 02 | 0.4118 | 0.6930 | 0.4328 | 0.4475 | 0.4963 | 0.6654 | 0.3201 | 0.4570 | 0.4571 | 0.8771 |
| 03 | 0.4116 | 0.6933 | 0.4331 | 0.4474 | 0.4963 | 0.6657 | 0.3201 | 0.4614 | 0.4572 | 0.8775 |
| 04 | 0.4152 | 0.6950 | 0.4352 | 0.4506 | 0.4990 | 0.6668 | 0.3219 | 0.4606 | 0.4587 | 0.8782 |
| 05 | 0.4116 | 0.6933 | 0.4325 | 0.4475 | 0.4959 | 0.6654 | 0.3198 | 0.4570 | 0.4573 | 0.8773 |
| 06 | 0.4115 | 0.6930 | 0.4322 | 0.4476 | 0.4959 | 0.6654 | 0.3201 | 0.4570 | 0.4570 | 0.8771 |
| 07 | 0.4112 | 0.6931 | 0.4319 | 0.4474 | 0.4963 | 0.6653 | 0.3198 | 0.4597 | 0.4572 | 0.8774 |
| 08 | 0.4150 | 0.6951 | 0.4343 | 0.4500 | 0.4990 | 0.6669 | 0.3222 | 0.4613 | 0.4587 | 0.8782 |
| 09 | 0.4115 | 0.6931 | 0.4325 | 0.4478 | 0.4966 | 0.6654 | 0.3198 | 0.4593 | 0.4573 | 0.8773 |
| 10 | 0.4115 | 0.6933 | 0.4326 | 0.4478 | 0.4959 | 0.6652 | 0.3201 | 0.4587 | 0.4571 | 0.8775 |
| 11 | 0.4114 | 0.6933 | 0.4320 | 0.4474 | 0.4956 | 0.6655 | 0.3201 | 0.4610 | 0.4573 | 0.8771 |
| 12 | 0.4150 | 0.6951 | 0.4343 | 0.4507 | 0.4990 | 0.6670 | 0.3219 | 0.4616 | 0.4587 | 0.8783 |
| 13 | 0.4147 | 0.6953 | 0.4345 | 0.4506 | 0.4990 | 0.6668 | 0.3222 | 0.4616 | 0.4588 | 0.8781 |
| 14 | 0.4148 | 0.6951 | 0.4349 | 0.4506 | 0.4997 | 0.6671 | 0.3219 | 0.4633 | 0.4588 | 0.8780 |
| 15 | 0.4148 | 0.6951 | 0.4346 | 0.4507 | 0.4997 | 0.6667 | 0.3222 | 0.4616 | 0.4583 | 0.8780 |
| 16 | 0.4148 | 0.6949 | 0.4342 | 0.4506 | 0.4997 | 0.6667 | 0.3222 | 0.4606 | 0.4587 | 0.8781 |
| 17 | 0.4115 | 0.6935 | 0.4332 | 0.4472 | 0.4966 | 0.6654 | 0.3201 | 0.4587 | 0.4569 | 0.8775 |
| 18 | 0.4150 | 0.6949 | 0.4345 | 0.4508 | 0.4993 | 0.6669 | 0.3226 | 0.4620 | 0.4587 | 0.8781 |
| 19 | 0.4116 | 0.6931 | 0.4323 | 0.4476 | 0.4963 | 0.6654 | 0.3201 | 0.4583 | 0.4571 | 0.8774 |
| 20 | 0.4149 | 0.6948 | 0.4348 | 0.4503 | 0.4990 | 0.6669 | 0.3219 | 0.4640 | 0.4588 | 0.8781 |
| 21 | 0.4147 | 0.6950 | 0.4350 | 0.4504 | 0.4993 | 0.6665 | 0.3222 | 0.4616 | 0.4587 | 0.8778 |
| 22 | 0.4149 | 0.6950 | 0.4342 | 0.4500 | 0.4987 | 0.6670 | 0.3222 | 0.4606 | 0.4587 | 0.8778 |
| 23 | 0.4150 | 0.6949 | 0.4343 | 0.4510 | 0.4990 | 0.6673 | 0.3219 | 0.4609 | 0.4585 | 0.8780 |

**Table 17.** (*Continued.*)

| run | stimulants | spices | alcoholic beverages | meat | offals | animal fats | milk—excluding butter | eggs | fish; seafood | aquatic products |
|---|---|---|---|---|---|---|---|---|---|---|
| 24 | 0.4146 | 0.6952 | 0.4348 | 0.4504 | 0.4997 | 0.6671 | 0.3219 | 0.4593 | 0.4586 | 0.8782 |
| 25 | 0.4117 | 0.6932 | 0.4325 | 0.4480 | 0.4963 | 0.6654 | 0.3198 | 0.4587 | 0.4572 | 0.8771 |
| 26 | 0.4115 | 0.6930 | 0.4322 | 0.4474 | 0.4953 | 0.6656 | 0.3198 | 0.4580 | 0.4572 | 0.8773 |
| 27 | 0.4147 | 0.6951 | 0.4343 | 0.4505 | 0.4990 | 0.6670 | 0.3226 | 0.4616 | 0.4586 | 0.8781 |
| 28 | 0.4148 | 0.6950 | 0.4343 | 0.4506 | 0.4990 | 0.6670 | 0.3222 | 0.4626 | 0.4586 | 0.8779 |
| 29 | 0.4148 | 0.6949 | 0.4348 | 0.4502 | 0.4990 | 0.6672 | 0.3219 | 0.4582 | 0.4584 | 0.8780 |
| 30 | 0.4117 | 0.6932 | 0.4320 | 0.4472 | 0.4959 | 0.6657 | 0.3198 | 0.4577 | 0.4569 | 0.8774 |
| mean | 0.4134 | 0.6942 | 0.4336 | 0.4492 | 0.4978 | 0.6663 | 0.3212 | 0.4601 | 0.4580 | 0.8777 |
| s.d. | 0.0017 | 0.0009 | 0.0011 | 0.0015 | 0.0016 | 0.0008 | 0.0011 | 0.0019 | 0.0008 | 0.0004 |

**Table 18.** The matching rate of the calibrated model for trade *volumes* for different regions (AFR: sub-Saharan Africa; CPA: centrally planned Asia; EUR: Europe; FSU: Former Soviet Union; LAM: Latin America; MEA: the Middle East/North Africa; NAM: North America; PAO: Pacific OECD; PAS: Pacific Asia; SAS: South Asia).

| run | AFR | CPA | EUR | FSU | LAM | MEA | NAM | PAO | PAS | SAS |
|---|---|---|---|---|---|---|---|---|---|---|
| 01 | 0.6254 | 0.5470 | 0.4337 | 0.5282 | 0.5956 | 0.4674 | 0.3272 | 0.4353 | 0.5559 | 0.5351 |
| 02 | 0.6252 | 0.5473 | 0.4338 | 0.5284 | 0.5959 | 0.4674 | 0.3257 | 0.4351 | 0.5562 | 0.5345 |
| 03 | 0.6254 | 0.5474 | 0.4338 | 0.5284 | 0.5956 | 0.4676 | 0.3263 | 0.4355 | 0.5566 | 0.5347 |
| 04 | 0.6253 | 0.5476 | 0.4338 | 0.5286 | 0.5955 | 0.4675 | 0.3251 | 0.4349 | 0.5565 | 0.5349 |
| 05 | 0.6253 | 0.5470 | 0.4338 | 0.5282 | 0.5957 | 0.4673 | 0.3251 | 0.4353 | 0.5565 | 0.5347 |
| 06 | 0.6252 | 0.5467 | 0.4337 | 0.5279 | 0.5951 | 0.4673 | 0.3263 | 0.4351 | 0.5566 | 0.5347 |
| 07 | 0.6251 | 0.5476 | 0.4337 | 0.5287 | 0.5955 | 0.4675 | 0.3263 | 0.4349 | 0.5563 | 0.5348 |
| 08 | 0.6252 | 0.5476 | 0.4338 | 0.5283 | 0.5956 | 0.4674 | 0.3257 | 0.4357 | 0.5559 | 0.5353 |
| 09 | 0.6253 | 0.5464 | 0.4340 | 0.5287 | 0.5957 | 0.4677 | 0.3260 | 0.4351 | 0.5562 | 0.5346 |
| 10 | 0.6254 | 0.5477 | 0.4337 | 0.5283 | 0.5958 | 0.4673 | 0.3260 | 0.4349 | 0.5567 | 0.5351 |
| 11 | 0.6251 | 0.5475 | 0.4337 | 0.5288 | 0.5953 | 0.4673 | 0.3263 | 0.4349 | 0.5562 | 0.5343 |
| 12 | 0.6252 | 0.5475 | 0.4338 | 0.5287 | 0.5956 | 0.4675 | 0.3263 | 0.4353 | 0.5566 | 0.5347 |
| 13 | 0.6254 | 0.5470 | 0.4338 | 0.5280 | 0.5955 | 0.4674 | 0.3266 | 0.4347 | 0.5560 | 0.5348 |
| 14 | 0.6254 | 0.5474 | 0.4338 | 0.5290 | 0.5955 | 0.4675 | 0.3263 | 0.4353 | 0.5565 | 0.5351 |
| 15 | 0.6254 | 0.5480 | 0.4337 | 0.5284 | 0.5957 | 0.4674 | 0.3260 | 0.4349 | 0.5560 | 0.5348 |
| 16 | 0.6254 | 0.5475 | 0.4338 | 0.5284 | 0.5957 | 0.4676 | 0.3257 | 0.4353 | 0.5559 | 0.5348 |
| 17 | 0.6250 | 0.5470 | 0.4338 | 0.5290 | 0.5955 | 0.4677 | 0.3263 | 0.4355 | 0.5562 | 0.5348 |
| 18 | 0.6254 | 0.5470 | 0.4338 | 0.5286 | 0.5957 | 0.4674 | 0.3260 | 0.4347 | 0.5562 | 0.5355 |
| 19 | 0.6252 | 0.5468 | 0.4338 | 0.5289 | 0.5956 | 0.4676 | 0.3260 | 0.4349 | 0.5560 | 0.5347 |
| 20 | 0.6255 | 0.5468 | 0.4336 | 0.5286 | 0.5959 | 0.4672 | 0.3254 | 0.4351 | 0.5568 | 0.5350 |
| 21 | 0.6253 | 0.5470 | 0.4336 | 0.5287 | 0.5953 | 0.4673 | 0.3257 | 0.4347 | 0.5565 | 0.5350 |
| 22 | 0.6253 | 0.5469 | 0.4337 | 0.5287 | 0.5954 | 0.4674 | 0.3254 | 0.4351 | 0.5561 | 0.5351 |

(*Continued.*)

**Table 18.** (*Continued.*)

| run | AFR | CPA | EUR | FSU | LAM | MEA | NAM | PAO | PAS | SAS |
|---|---|---|---|---|---|---|---|---|---|---|
| 23 | 0.6254 | 0.5478 | 0.4338 | 0.5287 | 0.5955 | 0.4675 | 0.3260 | 0.4345 | 0.5563 | 0.5350 |
| 24 | 0.6253 | 0.5478 | 0.4338 | 0.5288 | 0.5955 | 0.4674 | 0.3263 | 0.4353 | 0.5564 | 0.5348 |
| 25 | 0.6252 | 0.5474 | 0.4338 | 0.5282 | 0.5958 | 0.4675 | 0.3260 | 0.4349 | 0.5564 | 0.5347 |
| 26 | 0.6252 | 0.5469 | 0.4337 | 0.5284 | 0.5953 | 0.4676 | 0.3260 | 0.4353 | 0.5565 | 0.5348 |
| 27 | 0.6254 | 0.5480 | 0.4338 | 0.5283 | 0.5959 | 0.4677 | 0.3263 | 0.4357 | 0.5561 | 0.5348 |
| 28 | 0.6253 | 0.5471 | 0.4337 | 0.5289 | 0.5953 | 0.4674 | 0.3257 | 0.4357 | 0.5563 | 0.5351 |
| 29 | 0.6252 | 0.5473 | 0.4340 | 0.5283 | 0.5954 | 0.4674 | 0.3257 | 0.4355 | 0.5558 | 0.5348 |
| 30 | 0.6252 | 0.5477 | 0.4337 | 0.5280 | 0.5951 | 0.4673 | 0.3257 | 0.4355 | 0.5567 | 0.5350 |
| mean | 0.6253 | 0.5473 | 0.4338 | 0.5285 | 0.5955 | 0.4674 | 0.3260 | 0.4351 | 0.5563 | 0.5349 |
| s.d. | 0.0001 | 0.0004 | 0.0001 | 0.0003 | 0.0002 | 0.0001 | 0.0004 | 0.0003 | 0.0003 | 0.0002 |

**Table 19.** The matching rate of the calibrated model for trade *volumes* for different years.

| run | year2008 | year2009 | year2010 | year2011 | year2012 | year2013 |
|---|---|---|---|---|---|---|
| 01 | 0.5697 | 0.5649 | 0.5534 | 0.5426 | 0.5336 | 0.5289 |
| 02 | 0.5681 | 0.5633 | 0.5517 | 0.5410 | 0.5313 | 0.5267 |
| 03 | 0.5680 | 0.5635 | 0.5516 | 0.5411 | 0.5313 | 0.5269 |
| 04 | 0.5697 | 0.5650 | 0.5533 | 0.5428 | 0.5334 | 0.5288 |
| 05 | 0.5681 | 0.5631 | 0.5517 | 0.5409 | 0.5315 | 0.5267 |
| 06 | 0.5678 | 0.5631 | 0.5516 | 0.5405 | 0.5314 | 0.5265 |
| 07 | 0.5678 | 0.5633 | 0.5516 | 0.5409 | 0.5315 | 0.5266 |
| 08 | 0.5695 | 0.5650 | 0.5534 | 0.5427 | 0.5335 | 0.5287 |
| 09 | 0.5681 | 0.5633 | 0.5517 | 0.5412 | 0.5316 | 0.5267 |
| 10 | 0.5680 | 0.5634 | 0.5517 | 0.5411 | 0.5314 | 0.5270 |
| 11 | 0.5678 | 0.5632 | 0.5516 | 0.5407 | 0.5312 | 0.5268 |
| 12 | 0.5696 | 0.5650 | 0.5536 | 0.5428 | 0.5336 | 0.5288 |
| 13 | 0.5699 | 0.5647 | 0.5533 | 0.5426 | 0.5335 | 0.5288 |
| 14 | 0.5698 | 0.5652 | 0.5534 | 0.5428 | 0.5335 | 0.5290 |
| 15 | 0.5697 | 0.5649 | 0.5535 | 0.5428 | 0.5335 | 0.5287 |
| 16 | 0.5697 | 0.5649 | 0.5533 | 0.5429 | 0.5334 | 0.5289 |
| 17 | 0.5678 | 0.5632 | 0.5517 | 0.5409 | 0.5315 | 0.5267 |
| 18 | 0.5698 | 0.5651 | 0.5532 | 0.5431 | 0.5335 | 0.5288 |
| 19 | 0.5678 | 0.5632 | 0.5517 | 0.5410 | 0.5316 | 0.5267 |
| 20 | 0.5699 | 0.5650 | 0.5534 | 0.5430 | 0.5335 | 0.5287 |
| 21 | 0.5698 | 0.5649 | 0.5532 | 0.5425 | 0.5335 | 0.5287 |
| 22 | 0.5697 | 0.5651 | 0.5533 | 0.5426 | 0.5333 | 0.5287 |
| 23 | 0.5697 | 0.5652 | 0.5532 | 0.5428 | 0.5335 | 0.5287 |
| 24 | 0.5697 | 0.5650 | 0.5533 | 0.5427 | 0.5336 | 0.5290 |
| 25 | 0.5678 | 0.5631 | 0.5518 | 0.5411 | 0.5315 | 0.5269 |
| 26 | 0.5680 | 0.5633 | 0.5516 | 0.5408 | 0.5315 | 0.5264 |
| 27 | 0.5697 | 0.5652 | 0.5535 | 0.5428 | 0.5337 | 0.5288 |
| 28 | 0.5697 | 0.5649 | 0.5533 | 0.5427 | 0.5334 | 0.5287 |
| 29 | 0.5696 | 0.5649 | 0.5532 | 0.5424 | 0.5335 | 0.5289 |
| 30 | 0.5678 | 0.5635 | 0.5514 | 0.5407 | 0.5315 | 0.5265 |
| mean | 0.5689 | 0.5642 | 0.5526 | 0.5419 | 0.5326 | 0.5279 |
| s.d. | 0.0009 | 0.0009 | 0.0009 | 0.0009 | 0.0010 | 0.0011 |

**Table 20.** The matching rate of the calibrated model for trade *partners* for different food categories.

| run | overall | cereals | starchy roots | sugar crops | pulses | treenuts | oilcrops | vegetables | fruits | stimulants | spices |
|---|---|---|---|---|---|---|---|---|---|---|---|
| 01 | 0.6727 | 0.6890 | 0.3892 | 0.7141 | 0.5362 | 0.6628 | 0.9020 | 0.7090 | 0.6769 | 0.9199 | 0.8655 |
| 02 | 0.6728 | 0.6900 | 0.3877 | 0.7146 | 0.5394 | 0.6644 | 0.9023 | 0.7115 | 0.6750 | 0.9167 | 0.8739 |
| 03 | 0.6728 | 0.6883 | 0.3914 | 0.7138 | 0.5379 | 0.6636 | 0.9013 | 0.7113 | 0.6765 | 0.9187 | 0.8917 |
| 04 | 0.6728 | 0.6894 | 0.3881 | 0.7141 | 0.5407 | 0.6638 | 0.9017 | 0.7055 | 0.6763 | 0.9192 | 0.8655 |
| 05 | 0.6732 | 0.6886 | 0.3897 | 0.7143 | 0.5373 | 0.6677 | 0.9022 | 0.7083 | 0.6765 | 0.9191 | 0.8655 |
| 06 | 0.6729 | 0.6882 | 0.3871 | 0.7136 | 0.5397 | 0.6669 | 0.9014 | 0.7124 | 0.6770 | 0.9178 | 0.8750 |
| 07 | 0.6732 | 0.6908 | 0.3883 | 0.7130 | 0.5378 | 0.6638 | 0.9019 | 0.7113 | 0.6771 | 0.9153 | 0.8824 |
| 08 | 0.6733 | 0.6887 | 0.3876 | 0.7136 | 0.5394 | 0.6654 | 0.9025 | 0.7158 | 0.6770 | 0.9187 | 0.8729 |
| 09 | 0.6733 | 0.6885 | 0.3905 | 0.7148 | 0.5398 | 0.6663 | 0.9023 | 0.7080 | 0.6760 | 0.9199 | 0.8833 |
| 10 | 0.6742 | 0.6891 | 0.3903 | 0.7144 | 0.5389 | 0.6667 | 0.9026 | 0.7180 | 0.6778 | 0.9175 | 0.8833 |
| 11 | 0.6725 | 0.6888 | 0.3874 | 0.7137 | 0.5389 | 0.6648 | 0.9019 | 0.7145 | 0.6743 | 0.9177 | 0.8908 |
| 12 | 0.6726 | 0.6892 | 0.3851 | 0.7132 | 0.5382 | 0.6640 | 0.9013 | 0.7122 | 0.6777 | 0.9157 | 0.8824 |
| 13 | 0.6728 | 0.6894 | 0.3865 | 0.7138 | 0.5396 | 0.6638 | 0.9023 | 0.7134 | 0.6768 | 0.9171 | 0.8750 |
| 14 | 0.6729 | 0.6891 | 0.3865 | 0.7145 | 0.5410 | 0.6640 | 0.9019 | 0.7127 | 0.6782 | 0.9135 | 0.8908 |
| 15 | 0.6732 | 0.6886 | 0.3859 | 0.7123 | 0.5405 | 0.6652 | 0.9023 | 0.7163 | 0.6783 | 0.9165 | 0.8824 |
| 16 | 0.6727 | 0.6894 | 0.3871 | 0.7133 | 0.5397 | 0.6654 | 0.9020 | 0.7074 | 0.6783 | 0.9124 | 0.8750 |
| 17 | 0.6733 | 0.6889 | 0.3942 | 0.7145 | 0.5398 | 0.6634 | 0.9022 | 0.7097 | 0.6779 | 0.9147 | 0.8833 |
| 18 | 0.6737 | 0.6889 | 0.3872 | 0.7139 | 0.5414 | 0.6668 | 0.9019 | 0.7106 | 0.6788 | 0.9172 | 0.8824 |
| 19 | 0.6728 | 0.6891 | 0.3872 | 0.7138 | 0.5383 | 0.6648 | 0.9026 | 0.7116 | 0.6754 | 0.9206 | 0.8678 |
| 20 | 0.6728 | 0.6891 | 0.3876 | 0.7139 | 0.5384 | 0.6610 | 0.9022 | 0.7102 | 0.6780 | 0.9181 | 0.8843 |
| 21 | 0.6737 | 0.6900 | 0.3880 | 0.7142 | 0.5395 | 0.6680 | 0.9017 | 0.7097 | 0.6777 | 0.9151 | 0.8667 |
| 22 | 0.6725 | 0.6886 | 0.3857 | 0.7130 | 0.5385 | 0.6649 | 0.9020 | 0.7130 | 0.6763 | 0.9173 | 0.8655 |
| 23 | 0.6724 | 0.6889 | 0.3881 | 0.7130 | 0.5400 | 0.6649 | 0.9022 | 0.7125 | 0.6769 | 0.9106 | 0.8667 |

(Continued.)

**Table 20.** (*Continued.*)

| run | overall | cereals | starchy roots | sugar crops | pulses | treenuts | oilcrops | vegetables | fruits | stimulants | spices |
|---|---|---|---|---|---|---|---|---|---|---|---|
| 24 | 0.6732 | 0.6892 | 0.3865 | 0.7131 | 0.5386 | 0.6636 | 0.9017 | 0.7167 | 0.6783 | 0.9158 | 0.8824 |
| 25 | 0.6735 | 0.6902 | 0.3902 | 0.7140 | 0.5389 | 0.6653 | 0.9017 | 0.7093 | 0.6781 | 0.9144 | 0.8655 |
| 26 | 0.6736 | 0.6888 | 0.3892 | 0.7139 | 0.5372 | 0.6683 | 0.9019 | 0.7158 | 0.6774 | 0.9152 | 0.8655 |
| 27 | 0.6719 | 0.6884 | 0.3862 | 0.7143 | 0.5400 | 0.6636 | 0.9020 | 0.7035 | 0.6759 | 0.9172 | 0.8824 |
| 28 | 0.6732 | 0.6887 | 0.3861 | 0.7137 | 0.5388 | 0.6659 | 0.9016 | 0.7178 | 0.6766 | 0.9176 | 0.8843 |
| 29 | 0.6737 | 0.6900 | 0.3872 | 0.7146 | 0.5392 | 0.6657 | 0.9020 | 0.7168 | 0.6776 | 0.9153 | 0.8833 |
| 30 | 0.6735 | 0.6896 | 0.3862 | 0.7145 | 0.5388 | 0.6668 | 0.9022 | 0.7186 | 0.6764 | 0.9159 | 0.8739 |
| mean | 0.6731 | 0.6891 | 0.3879 | 0.7139 | 0.5391 | 0.6651 | 0.9020 | 0.7121 | 0.6770 | 0.9167 | 0.8770 |
| s.d. | 0.0005 | 0.0006 | 0.0020 | 0.0006 | 0.0012 | 0.0017 | 0.0004 | 0.0038 | 0.0011 | 0.0023 | 0.0086 |

**Table 21.** The matching rate of the calibrated model for trade *partners* for different years.

| run | year2008 | year2009 | year2010 | year2011 | year2012 | year2013 |
|---|---|---|---|---|---|---|
| 01 | 0.6888 | 0.6803 | 0.6578 | 0.6698 | 0.6710 | 0.6685 |
| 02 | 0.6899 | 0.6802 | 0.6587 | 0.6696 | 0.6693 | 0.6691 |
| 03 | 0.6896 | 0.6816 | 0.6572 | 0.6689 | 0.6705 | 0.6692 |
| 04 | 0.6894 | 0.6818 | 0.6576 | 0.6679 | 0.6718 | 0.6683 |
| 05 | 0.6901 | 0.6832 | 0.6592 | 0.6697 | 0.6692 | 0.6678 |
| 06 | 0.6889 | 0.6818 | 0.6587 | 0.6706 | 0.6690 | 0.6685 |
| 07 | 0.6892 | 0.6818 | 0.6584 | 0.6698 | 0.6694 | 0.6704 |
| 08 | 0.6897 | 0.6819 | 0.6596 | 0.6693 | 0.6699 | 0.6689 |
| 09 | 0.6878 | 0.6837 | 0.6577 | 0.6697 | 0.6713 | 0.6695 |
| 10 | 0.6900 | 0.6835 | 0.6592 | 0.6707 | 0.6716 | 0.6699 |
| 11 | 0.6886 | 0.6807 | 0.6586 | 0.6708 | 0.6684 | 0.6679 |
| 12 | 0.6895 | 0.6805 | 0.6581 | 0.6712 | 0.6682 | 0.6683 |
| 13 | 0.6890 | 0.6811 | 0.6578 | 0.6684 | 0.6703 | 0.6701 |
| 14 | 0.6903 | 0.6809 | 0.6570 | 0.6713 | 0.6698 | 0.6680 |
| 15 | 0.6904 | 0.6805 | 0.6592 | 0.6704 | 0.6705 | 0.6683 |
| 16 | 0.6893 | 0.6811 | 0.6567 | 0.6713 | 0.6689 | 0.6689 |
| 17 | 0.6896 | 0.6817 | 0.6601 | 0.6698 | 0.6697 | 0.6687 |
| 18 | 0.6907 | 0.6816 | 0.6595 | 0.6714 | 0.6698 | 0.6689 |
| 19 | 0.6884 | 0.6807 | 0.6593 | 0.6696 | 0.6692 | 0.6694 |
| 20 | 0.6907 | 0.6805 | 0.6593 | 0.6692 | 0.6693 | 0.6676 |
| 21 | 0.6888 | 0.6808 | 0.6594 | 0.6717 | 0.6708 | 0.6706 |
| 22 | 0.6892 | 0.6819 | 0.6567 | 0.6711 | 0.6674 | 0.6688 |
| 23 | 0.6891 | 0.6806 | 0.6578 | 0.6684 | 0.6698 | 0.6687 |
| 24 | 0.6892 | 0.6814 | 0.6581 | 0.6717 | 0.6694 | 0.6691 |
| 25 | 0.6903 | 0.6823 | 0.6593 | 0.6708 | 0.6690 | 0.6690 |
| 26 | 0.6901 | 0.6821 | 0.6586 | 0.6713 | 0.6703 | 0.6691 |
| 27 | 0.6895 | 0.6805 | 0.6571 | 0.6698 | 0.6671 | 0.6675 |
| 28 | 0.6895 | 0.6814 | 0.6589 | 0.6702 | 0.6695 | 0.6696 |
| 29 | 0.6912 | 0.6825 | 0.6573 | 0.6721 | 0.6694 | 0.6696 |
| 30 | 0.6889 | 0.6818 | 0.6591 | 0.6716 | 0.6702 | 0.6689 |
| mean | 0.6895 | 0.6815 | 0.6584 | 0.6703 | 0.6697 | 0.6689 |
| s.d. | 0.0008 | 0.0009 | 0.0010 | 0.0011 | 0.0011 | 0.0008 |

**Table 22.** Nutrition intake per capita of countries in 2013.

| country | calories | protein | fat | vitamin C | vitamin A | folate | iron | zinc | thiamin | riboflavin | niacin | saturated FA |
|---|---|---|---|---|---|---|---|---|---|---|---|---|
| Afghanistan | 2144 | 67 | 36 | 35 | 369 | 237 | 12.4 | 8.8 | 1.3 | 1.2 | 14.3 | 12.5 |
| Albania | 3545 | 112 | 110 | 190 | 1295 | 484 | 19.7 | 14.5 | 1.5 | 3.9 | 19.1 | 38.7 |
| Algeria | 3615 | 101 | 83 | 147 | 805 | 487 | 17.7 | 11.5 | 1.6 | 2.3 | 17.4 | 25.3 |
| Angola | 2330 | 52 | 48 | 159 | 728 | 390 | 11.4 | 7.7 | 1.6 | 0.7 | 16.4 | 14.6 |
| Antigua and Barbuda | 1847 | 71 | 45 | 138 | 494 | 578 | 13.8 | 7.5 | 1.8 | 1.7 | 24.8 | 16.9 |
| Argentina | 3253 | 97 | 119 | 82 | 769 | 901 | 21.2 | 12.1 | 2.9 | 3.1 | 32.9 | 38.5 |
| Armenia | 2904 | 89 | 96 | 244 | 1474 | 456 | 18.0 | 10.6 | 1.3 | 2.7 | 15.9 | 32.3 |
| Australia | 3413 | 98 | 171 | 87 | 1058 | 642 | 19.3 | 12.7 | 2.3 | 3.0 | 29.9 | 55.3 |
| Austria | 4027 | 98 | 206 | 130 | 980 | 335 | 14.2 | 12.3 | 1.8 | 2.9 | 20.4 | 69.5 |
| Azerbaijan | 3023 | 93 | 57 | 125 | 799 | 360 | 13.9 | 10.0 | 1.2 | 2.0 | 15.1 | 21.6 |
| Bahamas | 2705 | 84 | 106 | 251 | 868 | 560 | 20.1 | 11.2 | 2.3 | 2.3 | 28.1 | 38.5 |
| Bangladesh | 2387 | 53 | 29 | 40 | 332 | 188 | 19.3 | 8.5 | 2.3 | 0.9 | 19.3 | 9.3 |
| Barbados | 2975 | 88 | 92 | 111 | 702 | 768 | 19.2 | 9.9 | 2.5 | 2.4 | 30.6 | 33.3 |
| Belarus | 3308 | 91 | 140 | 136 | 988 | 349 | 17.3 | 12.4 | 1.9 | 2.1 | 22.6 | 41.0 |
| Belgium | 3976 | 94 | 191 | 118 | 1973 | 341 | 14.1 | 11.3 | 1.5 | 2.7 | 17.9 | 74.3 |
| Belize | 2774 | 76 | 81 | 144 | 386 | 728 | 19.0 | 10.0 | 2.8 | 2.0 | 24.7 | 24.6 |
| Benin | 2917 | 70 | 61 | 182 | 680 | 516 | 20.5 | 11.5 | 2.7 | 1.0 | 26.2 | 14.8 |
| Bermuda | 2148 | 83 | 67 | 237 | 912 | 323 | 13.1 | 9.4 | 1.3 | 1.9 | 19.0 | 24.7 |
| Bolivia (Plurinational State of) | 2559 | 75 | 73 | 75 | 369 | 591 | 18.9 | 10.5 | 2.5 | 1.8 | 27.0 | 21.8 |
| Bosnia and Herzegovina | 3129 | 94 | 80 | 197 | 1411 | 502 | 22.9 | 13.5 | 2.1 | 2.8 | 22.4 | 25.2 |
| Botswana | 2381 | 64 | 66 | 84 | 514 | 293 | 15.2 | 9.5 | 1.3 | 1.5 | 15.6 | 17.7 |
| Brazil | 3503 | 100 | 132 | 96 | 572 | 791 | 22.3 | 13.2 | 2.7 | 2.5 | 29.6 | 48.9 |

(*Continued.*)

**Table 22.** (*Continued.*)

| country | calories | protein | fat | vitamin C | vitamin A | folate | iron | zinc | thiamin | riboflavin | niacin | saturated FA |
|---|---|---|---|---|---|---|---|---|---|---|---|---|
| Brunei Darussalam | 3025 | 94 | 93 | 114 | 909 | 278 | 19.6 | 12.4 | 1.8 | 2.0 | 26.5 | 31.9 |
| Bulgaria | 2907 | 79 | 108 | 76 | 603 | 262 | 11.6 | 9.1 | 1.3 | 2.0 | 15.9 | 32.4 |
| Burkina Faso | 3383 | 91 | 71 | 23 | 367 | 596 | 25.0 | 14.6 | 2.9 | 1.6 | 32.0 | 16.4 |
| Cabo Verde | 2565 | 73 | 68 | 120 | 794 | 667 | 20.1 | 10.1 | 2.6 | 2.1 | 24.4 | 23.7 |
| Cambodia | 2543 | 60 | 34 | 57 | 338 | 248 | 20.5 | 9.3 | 2.4 | 1.0 | 21.1 | 10.5 |
| Cameroon | 3016 | 77 | 66 | 159 | 807 | 827 | 24.6 | 12.0 | 2.8 | 1.3 | 27.3 | 16.7 |
| Canada | 3728 | 99 | 177 | 121 | 839 | 882 | 22.3 | 13.3 | 2.9 | 3.1 | 31.2 | 54.4 |
| Central African Republic | 2116 | 46 | 65 | 125 | 323 | 358 | 11.9 | 8.4 | 1.7 | 0.7 | 14.5 | 15.0 |
| Chad | 2328 | 72 | 54 | 32 | 185 | 428 | 19.5 | 12.5 | 2.1 | 1.1 | 23.5 | 10.7 |
| Chile | 3228 | 94 | 99 | 85 | 506 | 980 | 22.9 | 11.3 | 3.4 | 2.8 | 34.3 | 33.6 |
| China; mainland | 2750 | 90 | 86 | 275 | 1687 | 576 | 26.3 | 12.3 | 2.6 | 2.1 | 23.8 | 25.3 |
| Colombia | 2984 | 68 | 90 | 127 | 768 | 487 | 15.9 | 9.2 | 1.9 | 1.8 | 21.6 | 33.2 |
| Congo | 2306 | 48 | 47 | 164 | 583 | 583 | 12.9 | 6.4 | 2.1 | 0.9 | 20.9 | 14.4 |
| Costa Rica | 3119 | 78 | 108 | 116 | 808 | 601 | 17.6 | 10.1 | 2.2 | 2.4 | 22.4 | 40.5 |
| Croatia | 3234 | 85 | 137 | 85 | 727 | 256 | 11.9 | 10.3 | 1.4 | 2.7 | 15.9 | 47.1 |
| Cuba | 3505 | 93 | 75 | 183 | 915 | 979 | 28.0 | 12.5 | 3.4 | 2.4 | 29.2 | 26.3 |
| Cyprus | 2576 | 73 | 113 | 90 | 580 | 254 | 11.3 | 8.8 | 1.3 | 1.8 | 15.6 | 31.3 |
| Czechia | 3442 | 84 | 159 | 81 | 847 | 282 | 11.8 | 10.1 | 1.4 | 2.2 | 17.5 | 49.3 |
| Democratic People's Republic of Korea | 2213 | 60 | 38 | 128 | 683 | 411 | 21.2 | 10.0 | 2.3 | 1.1 | 18.8 | 9.3 |
| Denmark | 3733 | 106 | 154 | 123 | 1248 | 342 | 16.8 | 14.2 | 1.6 | 3.2 | 21.0 | 59.3 |
| Djibouti | 2585 | 65 | 55 | 67 | 687 | 1022 | 22.1 | 7.4 | 3.2 | 2.3 | 27.8 | 18.7 |
| Dominica | 2636 | 70 | 51 | 336 | 805 | 805 | 18.1 | 7.9 | 2.6 | 2.0 | 25.9 | 23.2 |
| Dominican Republic | 2750 | 61 | 95 | 167 | 780 | 546 | 16.4 | 7.8 | 1.8 | 1.4 | 20.8 | 37.3 |

(*Continued.*)

**Table 22.** (*Continued.*)

| country | calories | protein | fat | vitamin C | vitamin A | folate | iron | zinc | thiamin | riboflavin | niacin | saturated FA |
|---|---|---|---|---|---|---|---|---|---|---|---|---|
| Ecuador | 2483 | 64 | 103 | 63 | 784 | 441 | 13.9 | 8.2 | 1.9 | 1.8 | 20.3 | 38.7 |
| Egypt | 3247 | 91 | 57 | 128 | 713 | 387 | 20.4 | 11.4 | 1.9 | 1.7 | 21.0 | 16.9 |
| El Salvador | 2662 | 74 | 70 | 74 | 471 | 604 | 18.6 | 10.9 | 2.2 | 2.0 | 21.5 | 25.4 |
| Estonia | 3162 | 98 | 101 | 118 | 956 | 382 | 16.3 | 13.0 | 1.7 | 2.9 | 19.7 | 36.7 |
| Ethiopia | 2108 | 67 | 33 | 38 | 444 | 417 | 18.4 | 11.8 | 1.9 | 1.2 | 18.1 | 9.5 |
| Fiji | 3183 | 76 | 103 | 89 | 507 | 892 | 22.6 | 9.3 | 3.0 | 1.8 | 29.9 | 54.3 |
| Finland | 3343 | 109 | 144 | 103 | 1075 | 750 | 18.2 | 13.5 | 2.7 | 4.4 | 29.7 | 56.5 |
| France | 3628 | 102 | 168 | 127 | 1137 | 318 | 15.1 | 12.8 | 1.5 | 2.6 | 19.2 | 60.4 |
| French Polynesia | 2992 | 94 | 123 | 109 | 667 | 283 | 14.8 | 11.3 | 1.4 | 1.7 | 23.9 | 49.8 |
| Gabon | 2780 | 80 | 62 | 140 | 640 | 343 | 14.8 | 9.2 | 1.7 | 1.0 | 24.4 | 17.1 |
| Gambia | 2818 | 65 | 74 | 22 | 502 | 270 | 16.1 | 9.6 | 1.9 | 1.2 | 21.8 | 19.5 |
| Georgia | 3057 | 86 | 72 | 67 | 470 | 251 | 12.1 | 9.7 | 1.3 | 1.9 | 15.1 | 23.3 |
| Germany | 3787 | 95 | 181 | 100 | 999 | 299 | 12.8 | 12.0 | 1.6 | 2.9 | 19.3 | 63.8 |
| Ghana | 3179 | 64 | 52 | 272 | 676 | 641 | 20.7 | 10.5 | 2.6 | 0.9 | 25.0 | 17.1 |
| Greece | 3543 | 103 | 161 | 170 | 1039 | 422 | 17.4 | 12.8 | 1.6 | 2.9 | 19.0 | 44.2 |
| Grenada | 2149 | 68 | 49 | 131 | 425 | 665 | 15.2 | 7.1 | 1.9 | 1.8 | 23.7 | 21.6 |
| Guatemala | 2511 | 65 | 61 | 72 | 434 | 592 | 18.0 | 10.2 | 2.3 | 1.7 | 20.8 | 18.2 |
| Guinea | 2656 | 53 | 62 | 134 | 709 | 401 | 18.0 | 8.9 | 2.3 | 1.0 | 20.7 | 18.1 |
| Guinea-Bissau | 2364 | 46 | 65 | 56 | 395 | 216 | 15.3 | 7.4 | 1.9 | 0.8 | 16.6 | 22.0 |
| Guyana | 3039 | 86 | 70 | 98 | 588 | 738 | 24.8 | 11.2 | 2.8 | 2.4 | 30.2 | 35.8 |
| Haiti | 2010 | 46 | 49 | 83 | 677 | 474 | 14.0 | 7.2 | 1.8 | 0.8 | 14.6 | 15.2 |
| Honduras | 2550 | 62 | 72 | 67 | 733 | 297 | 12.8 | 9.4 | 1.5 | 1.4 | 15.7 | 25.1 |
| Hungary | 3197 | 77 | 156 | 81 | 876 | 255 | 11.0 | 8.9 | 1.2 | 1.9 | 14.9 | 51.2 |

(*Continued.*)

**Table 22.** (*Continued.*)

| country | calories | protein | fat | vitamin C | vitamin A | folate | iron | zinc | thiamin | riboflavin | niacin | saturated FA |
|---|---|---|---|---|---|---|---|---|---|---|---|---|
| Iceland | 2858 | 100 | 113 | 102 | 960 | 247 | 11.6 | 11.8 | 1.2 | 2.7 | 21.2 | 48.4 |
| India | 2543 | 62 | 55 | 91 | 568 | 393 | 17.0 | 8.3 | 1.6 | 1.3 | 14.2 | 19.7 |
| Indonesia | 2967 | 66 | 64 | 84 | 546 | 411 | 22.9 | 10.4 | 2.8 | 1.3 | 26.5 | 28.9 |
| Iran (Islamic Republic of) | 3478 | 99 | 82 | 217 | 998 | 554 | 20.9 | 10.8 | 1.8 | 1.9 | 19.7 | 21.9 |
| Iraq | 2546 | 65 | 60 | 85 | 631 | 1120 | 23.6 | 6.8 | 3.4 | 2.4 | 30.9 | 15.3 |
| Ireland | 3648 | 104 | 144 | 152 | 964 | 368 | 13.6 | 12.7 | 1.7 | 2.8 | 21.2 | 54.8 |
| Israel | 3775 | 121 | 164 | 159 | 1186 | 502 | 20.8 | 15.0 | 1.7 | 2.8 | 25.9 | 48.3 |
| Italy | 4010 | 109 | 189 | 146 | 1113 | 393 | 15.7 | 12.7 | 1.7 | 3.0 | 18.7 | 60.9 |
| Jamaica | 2796 | 77 | 83 | 159 | 1022 | 769 | 21.7 | 8.8 | 2.6 | 2.2 | 29.7 | 37.1 |
| Japan | 2525 | 75 | 94 | 90 | 712 | 314 | 17.3 | 10.1 | 1.6 | 1.8 | 17.7 | 26.4 |
| Jordan | 2740 | 70 | 87 | 84 | 763 | 319 | 12.1 | 7.5 | 1.1 | 1.3 | 13.6 | 24.5 |
| Kazakhstan | 3217 | 96 | 132 | 159 | 1152 | 927 | 23.7 | 12.4 | 2.9 | 3.5 | 30.1 | 38.5 |
| Kenya | 2423 | 66 | 52 | 86 | 756 | 645 | 19.8 | 10.7 | 2.4 | 1.8 | 21.4 | 18.4 |
| Kiribati | 2826 | 60 | 102 | 93 | 344 | 577 | 20.5 | 7.3 | 2.3 | 1.1 | 24.4 | 77.1 |
| Kuwait | 3310 | 102 | 112 | 173 | 1211 | 1029 | 29.2 | 12.9 | 3.2 | 3.2 | 35.9 | 36.2 |
| Kyrgyzstan | 2870 | 87 | 67 | 104 | 777 | 367 | 14.5 | 10.9 | 1.4 | 2.1 | 14.9 | 24.0 |
| Lao People's Democratic Republic | 2747 | 67 | 37 | 208 | 1111 | 430 | 28.0 | 11.1 | 3.0 | 1.8 | 26.5 | 9.2 |
| Latvia | 3277 | 88 | 140 | 107 | 1081 | 309 | 15.3 | 11.7 | 1.7 | 2.4 | 20.7 | 49.5 |
| Lebanon | 2935 | 79 | 99 | 147 | 695 | 416 | 15.3 | 9.8 | 1.2 | 1.9 | 15.2 | 25.6 |
| Lesotho | 2789 | 80 | 40 | 46 | 228 | 285 | 20.1 | 14.8 | 2.5 | 1.4 | 25.5 | 9.5 |
| Liberia | 2274 | 38 | 60 | 92 | 786 | 248 | 13.0 | 6.1 | 1.8 | 0.6 | 15.9 | 22.5 |
| Lithuania | 3643 | 111 | 127 | 98 | 995 | 351 | 13.7 | 12.7 | 1.8 | 3.1 | 23.3 | 48.6 |
| Luxembourg | 3383 | 103 | 123 | 174 | 954 | 341 | 16.4 | 14.3 | 1.7 | 3.5 | 23.1 | 45.1 |

(*Continued.*)

**Table 22.** (*Continued.*)

| country | calories | protein | fat | vitamin C | vitamin A | folate | iron | zinc | thiamin | riboflavin | niacin | saturated FA |
|---|---|---|---|---|---|---|---|---|---|---|---|---|
| Madagascar | 2080 | 42 | 23 | 102 | 378 | 222 | 13.5 | 7.4 | 1.9 | 0.7 | 15.8 | 8.6 |
| Malawi | 2628 | 70 | 46 | 99 | 220 | 493 | 19.8 | 13.3 | 2.7 | 1.2 | 24.9 | 9.9 |
| Malaysia | 3109 | 85 | 94 | 76 | 943 | 251 | 19.1 | 10.2 | 1.8 | 1.4 | 23.6 | 38.1 |
| Maldives | 2369 | 92 | 56 | 122 | 615 | 262 | 14.6 | 8.7 | 1.5 | 1.9 | 24.3 | 18.9 |
| Mali | 2901 | 87 | 64 | 61 | 601 | 582 | 24.4 | 14.2 | 2.8 | 1.9 | 27.6 | 18.3 |
| Malta | 3914 | 118 | 129 | 173 | 1044 | 476 | 18.7 | 13.9 | 1.9 | 2.8 | 22.9 | 42.4 |
| Mauritania | 3026 | 85 | 73 | 27 | 659 | 338 | 14.2 | 10.2 | 1.4 | 1.7 | 15.3 | 27.8 |
| Mauritius | 3217 | 90 | 101 | 77 | 541 | 345 | 15.1 | 9.7 | 1.4 | 1.6 | 19.2 | 31.6 |
| Mexico | 3295 | 93 | 104 | 103 | 787 | 618 | 21.9 | 14.4 | 2.8 | 2.5 | 29.2 | 34.0 |
| Mongolia | 2442 | 86 | 75 | 59 | 709 | 239 | 12.4 | 11.0 | 1.0 | 1.8 | 16.0 | 27.9 |
| Morocco | 3634 | 103 | 75 | 111 | 626 | 1503 | 33.4 | 12.4 | 4.8 | 3.5 | 45.6 | 20.9 |
| Mozambique | 2515 | 50 | 45 | 134 | 433 | 488 | 14.1 | 9.2 | 2.3 | 0.8 | 18.4 | 12.8 |
| Myanmar | 2746 | 78 | 71 | 86 | 587 | 377 | 21.2 | 10.5 | 2.4 | 1.3 | 24.2 | 21.0 |
| Namibia | 2375 | 62 | 54 | 87 | 358 | 399 | 15.3 | 8.6 | 1.6 | 1.2 | 17.3 | 14.6 |
| Nepal | 2896 | 76 | 59 | 144 | 723 | 694 | 30.1 | 12.6 | 3.3 | 1.9 | 28.5 | 16.5 |
| Netherlands | 3325 | 99 | 144 | 144 | 1031 | 310 | 12.4 | 12.0 | 1.6 | 2.9 | 18.9 | 52.6 |
| New Caledonia | 2777 | 80 | 117 | 119 | 795 | 242 | 11.5 | 9.2 | 1.2 | 1.7 | 18.3 | 44.4 |
| New Zealand | 3157 | 88 | 131 | 114 | 1004 | 325 | 14.5 | 11.3 | 1.4 | 2.0 | 20.3 | 48.0 |
| Nicaragua | 2746 | 71 | 66 | 33 | 400 | 590 | 18.2 | 10.8 | 2.4 | 1.7 | 22.1 | 21.7 |
| Niger | 2999 | 102 | 65 | 59 | 481 | 949 | 27.6 | 15.7 | 3.0 | 2.1 | 32.1 | 17.1 |
| Nigeria | 3158 | 72 | 65 | 182 | 675 | 703 | 23.0 | 11.4 | 2.9 | 1.4 | 25.5 | 20.8 |
| Norway | 3422 | 101 | 149 | 112 | 870 | 325 | 13.7 | 13.0 | 1.4 | 2.8 | 18.9 | 54.7 |
| Oman | 3266 | 88 | 89 | 161 | 1008 | 346 | 19.4 | 11.6 | 1.5 | 2.1 | 22.1 | 31.3 |

(*Continued.*)

**Table 22.** (Continued.)

| country | calories | protein | fat | vitamin C | vitamin A | folate | iron | zinc | thiamin | riboflavin | niacin | saturated FA |
|---|---|---|---|---|---|---|---|---|---|---|---|---|
| Pakistan | 2738 | 72 | 82 | 40 | 795 | 259 | 10.4 | 8.1 | 0.9 | 1.7 | 10.2 | 32.0 |
| Panama | 2900 | 76 | 91 | 62 | 389 | 517 | 18.8 | 10.2 | 2.4 | 1.8 | 26.0 | 31.1 |
| Paraguay | 3021 | 73 | 111 | 139 | 584 | 598 | 16.6 | 11.4 | 2.6 | 1.9 | 23.1 | 37.3 |
| Peru | 2609 | 71 | 54 | 133 | 882 | 739 | 23.0 | 10.0 | 2.8 | 1.9 | 26.7 | 18.8 |
| Philippines | 2710 | 63 | 56 | 109 | 487 | 375 | 19.9 | 9.1 | 2.6 | 1.4 | 25.1 | 24.7 |
| Poland | 3565 | 93 | 142 | 99 | 1005 | 312 | 12.6 | 11.0 | 1.7 | 2.3 | 20.5 | 53.0 |
| Portugal | 3648 | 104 | 165 | 153 | 1169 | 375 | 16.9 | 12.8 | 1.8 | 2.5 | 22.4 | 55.2 |
| Republic of Korea | 3176 | 87 | 115 | 164 | 1095 | 454 | 24.1 | 12.7 | 2.3 | 2.0 | 22.3 | 36.5 |
| Republic of Moldova | 2258 | 61 | 84 | 59 | 531 | 189 | 11.1 | 8.9 | 1.4 | 1.9 | 15.0 | 24.1 |
| Romania | 3601 | 105 | 120 | 152 | 1166 | 417 | 17.9 | 12.8 | 1.8 | 2.8 | 21.1 | 37.6 |
| Russian Federation | 3347 | 94 | 117 | 103 | 811 | 314 | 13.5 | 10.6 | 1.4 | 2.0 | 19.0 | 36.7 |
| Rwanda | 2357 | 63 | 22 | 206 | 1107 | 864 | 20.1 | 10.0 | 2.2 | 0.7 | 18.1 | 6.5 |
| Saint Kitts and Nevis | 2031 | 63 | 48 | 76 | 295 | 132 | 7.7 | 7.0 | 0.7 | 1.1 | 15.8 | 22.5 |
| Saint Lucia | 2780 | 93 | 79 | 83 | 567 | 316 | 11.6 | 9.0 | 1.4 | 1.4 | 21.8 | 36.7 |
| Saint Vincent and the Grenadines | 2796 | 89 | 53 | 145 | 540 | 370 | 13.7 | 9.1 | 1.6 | 1.3 | 22.6 | 18.2 |
| Samoa | 3412 | 86 | 176 | 152 | 529 | 337 | 18.5 | 11.3 | 1.2 | 1.1 | 23.4 | 111.4 |
| Sao Tome and Principe | 2523 | 55 | 82 | 74 | 143 | 304 | 14.5 | 7.7 | 1.0 | 0.6 | 14.2 | 63.0 |
| Saudi Arabia | 3118 | 84 | 104 | 85 | 1090 | 858 | 24.4 | 10.3 | 2.9 | 2.5 | 32.4 | 34.5 |
| Senegal | 2718 | 65 | 75 | 44 | 451 | 436 | 19.4 | 9.9 | 2.5 | 1.4 | 25.5 | 20.0 |
| Sierra Leone | 2358 | 51 | 55 | 115 | 812 | 383 | 15.9 | 7.7 | 2.1 | 0.8 | 19.2 | 19.0 |
| Slovakia | 3200 | 78 | 128 | 65 | 565 | 253 | 10.4 | 9.0 | 1.2 | 2.1 | 14.9 | 41.6 |
| Slovenia | 3308 | 95 | 139 | 96 | 934 | 305 | 14.7 | 12.1 | 1.5 | 2.8 | 19.0 | 51.7 |
| Solomon Islands | 2323 | 53 | 53 | 146 | 1605 | 623 | 19.5 | 8.0 | 2.3 | 0.8 | 19.0 | 35.5 |

(Continued.)

**44**

**Table 22.** (*Continued.*)

| country | calories | protein | fat | vitamin C | vitamin A | folate | iron | zinc | thiamin | riboflavin | niacin | saturated FA |
|---|---|---|---|---|---|---|---|---|---|---|---|---|
| South Africa | 3245 | 89 | 89 | 47 | 511 | 668 | 22.7 | 13.0 | 3.1 | 2.3 | 33.6 | 24.4 |
| Spain | 3376 | 96 | 169 | 100 | 745 | 334 | 14.3 | 11.6 | 1.6 | 2.1 | 20.0 | 46.2 |
| Sri Lanka | 2796 | 61 | 61 | 58 | 375 | 289 | 21.1 | 9.1 | 1.8 | 1.0 | 17.6 | 42.9 |
| Suriname | 2804 | 65 | 84 | 110 | 431 | 548 | 17.8 | 8.0 | 2.3 | 1.5 | 25.5 | 28.6 |
| Sweden | 3487 | 100 | 164 | 122 | 1335 | 311 | 12.5 | 12.6 | 1.5 | 3.2 | 18.5 | 64.1 |
| Switzerland | 3645 | 93 | 165 | 116 | 1145 | 305 | 12.6 | 11.4 | 1.3 | 3.0 | 16.2 | 59.3 |
| Tajikistan | 2200 | 68 | 50 | 96 | 526 | 297 | 14.9 | 9.4 | 1.4 | 1.3 | 15.9 | 13.0 |
| Thailand | 2786 | 61 | 58 | 120 | 625 | 245 | 21.5 | 8.8 | 2.0 | 1.2 | 19.6 | 21.4 |
| The former Yugoslav Republic of Macedonia | 3040 | 80 | 113 | 150 | 1097 | 396 | 16.8 | 10.1 | 1.5 | 2.6 | 16.6 | 31.1 |
| Togo | 2612 | 62 | 49 | 104 | 345 | 489 | 17.6 | 10.7 | 2.5 | 1.0 | 21.9 | 17.0 |
| Trinidad and Tobago | 3209 | 90 | 104 | 102 | 490 | 913 | 22.7 | 10.0 | 2.9 | 2.5 | 33.3 | 41.6 |
| Tunisia | 3601 | 104 | 100 | 191 | 1070 | 533 | 19.4 | 11.0 | 1.4 | 2.1 | 17.5 | 30.2 |
| Turkey | 3956 | 112 | 129 | 174 | 1178 | 582 | 20.4 | 13.1 | 1.8 | 2.7 | 19.2 | 38.4 |
| Turkmenistan | 2836 | 93 | 74 | 86 | 540 | 1442 | 27.7 | 10.6 | 4.2 | 3.8 | 39.2 | 25.8 |
| Uganda | 2314 | 56 | 52 | 103 | 835 | 586 | 15.4 | 9.1 | 1.9 | 1.0 | 16.9 | 13.9 |
| Ukraine | 3117 | 85 | 102 | 127 | 828 | 341 | 14.2 | 10.3 | 1.5 | 2.0 | 18.7 | 30.3 |
| United Arab Emirates | 3632 | 115 | 108 | 94 | 621 | 594 | 22.5 | 14.1 | 1.9 | 2.2 | 22.8 | 32.5 |
| United Kingdom | 3282 | 94 | 134 | 117 | 792 | 840 | 20.8 | 11.5 | 2.9 | 3.1 | 29.9 | 44.0 |
| United Republic of Tanzania | 2457 | 61 | 54 | 109 | 774 | 596 | 18.0 | 10.4 | 2.3 | 1.1 | 20.0 | 17.0 |
| United States of America | 3869 | 105 | 189 | 103 | 742 | 752 | 20.0 | 14.2 | 2.6 | 3.4 | 32.2 | 63.7 |
| Uruguay | 3384 | 101 | 114 | 83 | 800 | 1011 | 23.6 | 12.9 | 3.3 | 3.4 | 34.6 | 41.6 |
| Uzbekistan | 2758 | 85 | 72 | 177 | 988 | 393 | 15.3 | 9.7 | 1.2 | 2.1 | 13.9 | 23.7 |

(*Continued.*)

**Table 22.** (Continued.)

| country | calories | protein | fat | vitamin C | vitamin A | folate | iron | zinc | thiamin | riboflavin | niacin | saturated FA |
|---|---|---|---|---|---|---|---|---|---|---|---|---|
| Vanuatu | 2938 | 70 | 122 | 113 | 476 | 443 | 20.7 | 10.0 | 1.7 | 0.8 | 20.5 | 83.0 |
| Venezuela (Bolivarian Republic of) | 3040 | 82 | 104 | 87 | 607 | 597 | 18.3 | 11.2 | 2.5 | 2.2 | 27.8 | 34.1 |
| Viet Nam | 2823 | 80 | 71 | 155 | 885 | 451 | 27.4 | 11.7 | 3.1 | 1.7 | 27.1 | 24.3 |
| Yemen | 2190 | 58 | 44 | 46 | 465 | 891 | 19.4 | 6.7 | 2.7 | 2.0 | 25.8 | 14.5 |
| Zambia | 2144 | 57 | 47 | 60 | 373 | 286 | 15.5 | 10.5 | 1.9 | 1.1 | 17.8 | 10.8 |
| Zimbabwe | 2295 | 52 | 60 | 26 | 245 | 326 | 14.4 | 9.3 | 2.0 | 1.2 | 19.2 | 14.8 |

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
