## [Reviewer comments · Royal Society Open Science]

Review History

RSOS-201587.R0 (Original submission)

Review form: Reviewer 1 (Alvaro Iribarrem)

Is the manuscript scientifically sound in its present form?

Yes

Are the interpretations and conclusions justified by the results?

Yes

Is the language acceptable?

Yes

Do you have any ethical concerns with this paper?

No

Have you any concerns about statistical analyses in this paper?

No

Recommendation?

Accept with minor revision (please list in comments)

Comments to the Author(s)

I would like to congratulate the authors on their very interesting and well-written work. I have written down my suggestions, questions, and considerations as I was going through the text, which I hope will be found useful. That is, perhaps, not the best way to clarify what are the exact revisions I'd suggest the Editor to ask the authors to make. For clarity, I enumerate the specific actions needed below -- they refer to the comments (a) to (h) that follow.

1. Clarifications needed for comments (a) to (c) can be done on section 2.2.2. One way of doing it is suggested at the end of comment (c);
2. Regarding comment (d), a brief description of the structure of maximum amount of trade is needed. By structure I mean is it a constant universal value? Is it country specific? Food-category specific? The sensitivity analysis suggested is not needed if the authors provide further explanation of why it is not reasonable to expect this parameter to change significantly the behavior of the model;
3. To address my comment (e), clarification about the order by which food-categories are traded is needed. If the order is not changed/randomized between years, a few lines discussing why this wouldn't introduce a bias against the trade of the last categories, or why this is not important for the results and the objectives of the paper would be required;
4. Comments (f) to (h) require clarifications, as suggested in the respective comments;

In my understanding the work is scientifically sound and novel enough for publication. In its current writing the manuscript raises a few questions that need to be addressed in order to better reflect that. Including the additions in the four points above would make the scientific content in the manuscript described well-enough to be published.

Kind regards,
Alvaro

--- Comments ---

(a) page 11, paragraph starting at lines 33-34. It is not clear if the trade volume information is corrected for the first two elements (GDP, and proximity). The authors wrote that historic trade volume will reflect these non-GDP and non-geographic factors, but isn't it reasonable to assume that trade volume will also be affected by GDP and geographic factors? Assume a country A exports the same volume in a given commodity in 2000 to two different neighboring countries, B and C, both with similar distances, but B having a significantly larger GDP than C. Would this third element, trade relationship, be assigned the same value for that country exporting that commodity, or have different values taking in consideration the difference in GDP between the two trade partners? Shouldn't the trade-relationship element for the export of that commodity from A to C have a higher value than that from A to B in this case? If not corrected, then it would be important to discuss why this wouldn't introduce a significant double-counting type of bias in favor of larger GDP / shorter distances when combining this third element with the previous two.

(b) page 12, paragraph starting at lines 3-4. In here the reader is left assuming that the ranking and matching process is calibrated somehow to account for trade-offs between the variables mentioned in the paragraph (production and transport costs, willingness to pay, etc...). Does that process take into consideration the effects of the different variables differently? For example, up to which point (quantity?) production costs outweigh transport costs in the match-making

process? A few sentences in here explaining how such calibration allows the model to reproduce the dynamics of price implicitly would make the argument in this paragraph more convincing. As it is, the argument sounds a bit incomplete: the model includes all the information in the price, but does it do so in a way that emulates the price dynamics sufficiently well?

(c) page 12, paragraph starting at lines 15-16. The calibration described here is a valid way of addressing the questions above in an empirical way. As a suggestion, the authors would be able to address both issues above in a relatively short way by adding a sentence at the end of the paragraph at page 11, line 6 stating that the four elements are calibrated with real data, and then a few sentences in each of the two paragraphs I've mentioned before stating how the calibration addresses the issues, namely of the possible double-count bias of the third element (they don't need to be orthogonal, as long as their parameters are calibrated to match observed data), and of the price dynamics (the behavior is calibrated to match the observations, and therefore it should work at least as well as having a price variable explicitly). Regardless of the way they choose to address these questions, I think it would be useful to have it in this section (2.2.2.).

(d) page 13, paragraph starting at line 31. The paragraph does a good job of explaining why the addition of the maximum amount of trade is important, but leaves the reader wondering how this parameter is computed, and what are the uncertainties in the model's behavior caused by its addition. If this is described elsewhere, it is worth pointing it out. If not, it would be important to describe the structure of this parameter (is it a flat value for all countries? does it change over time, or over commodities?), and also to provide some form of sensitivity analysis regarding its parameterization. What would be a reasonable variation of this maximum amount? What would be a good metric to check the impact of such variation? By how much would that metric change under that variation? Could the literature help ground any of these answers?

(e) page 13, paragraph starting at line 49. Is the order with which the food categories are traded the same every year? If so, wouldn't it introduce a bias against the consumption of the latter categories in the order, since possibly all the macro and micro nutrients needed by a country could be met by the trade and production of the first categories? Would it be possible to randomize the order the categories are traded from year to year in NetLogo (or perhaps feeding a new randomized order into NetLogo each year)? And if so, can it be used to show that following a fixed order yields no significant difference on the final result?

(f) page 14, paragraph starting at line 3. Although the use of the word parameter implies that trade saturation is a quantity that can be set in the model, it is only much later in the text (section 4.1) that this parameter is fixed to different values, and its impact discussed. From this paragraph alone it could also be the case that the authors let the model run until market clearance, saving snapshots of each food category's trade round at fixed saturation values. A few words in here clarifying that saturation is a parameter in the sense that it can be fixed by the model might help some readers understand it more concretely in here, and later help in the reading of section 4.1.

(g) page 18, paragraph starting at lines 44-45. The authors' approach to calibrating their model is sufficient for the scope of the current work. It would be interesting to explore in a future work how the calibrated parameters in table 5 would vary across countries, or at least regions. The significant differences between the mean values in table 18 suggests the general parameterization done presently works better for some regions than some others. Also, the use of two metrics for finding the Pareto tradeoff between them is a valid approach, but it doesn't guarantee that the solutions are matching volume and partner together. The same goes for the validation. In the future the authors might want to consider a single metric defined by number of matches (right partner with a volume difference within the same 20% range) over the total amount of trades. This single metric would allow the authors to pick the best parameterization in the random set of 10,000, or to inform some sort of gradient-descent / markov-chain mechanism, which might even

converge before 10,000 runs (particularly if the authors use their current calibration as starting point). For the present work, a few lines discussing how these two caveats, namely lack of parameterization per country, and the uncontrolled correlations between trade volumes and partners, are not an issue for its stated purposes (as in page 6, paragraph starting at line 29) would make section 3.1, and subsequently section 3.2, better.

(h) page 22, paragraph starting at line 55. If indeed trade saturation is a parameter that can be fixed in the model, the calibration and validation sections don't discussed explicitly to which value it was set for the analyses there. From my reading, currently the model doesn't have an endogenous mechanism that stops trading from happening before market clearance. I understand it is beyond the scope of the current work, but wouldn't it be reasonable to assume that relaxing the constraint of market clearance would allow the model to provide better predictions? If trade saturation is indeed a parameter, why not include it in the calibration process? In other words, it would be important to discuss precisely how trade saturation works in the model, and if it is indeed a fixed parameter, a constraint so to speak, why the choice of 1, or whatever choice was made in the calibration and validation phases, is better than, say 0.5, or a value that yielded the best calibration with the observed data?

Review form: Reviewer 2

Is the manuscript scientifically sound in its present form?

Yes

Are the interpretations and conclusions justified by the results?

Yes

Is the language acceptable?

Yes

Do you have any ethical concerns with this paper?

No

Have you any concerns about statistical analyses in this paper?

No

Recommendation?

Accept with minor revision (please list in comments)

Comments to the Author(s)

Food and Nutrition Security under Global Trade

GENERAL

The paper provides a useful contribution to the current set of global trade models using countries as agents that trade based on 4 main factors or criteria that are identified based on analysis of historic data. Key challenges that the authors could address include, first, move most of the model development specificities into the appendix. The paper is very long and many readers will not be able to hang on for the ride; try to address, if feasible some of the already identified challenges, such as relating future trade only to history and not to changing trends in diets (linked empirically to changes in incomes and urbanization) in low-income countries, and in high-income countries, to reductions in per capita consumption of some food items, like milk, sugary drinks, and a few other commodities.

P 28, lines 26 notes that poor countries are penalized by trade. Other studies have found that poor countries always benefit nutritionally from trade, pretty much regardless of scenario.

The results on pages 23-25 could be tightened and presented differently. My black and white printout showed nothing, also which trade saturation picture comes closest to reality?

The paper should take a bit more care to adequately differentiate the terms food security and nutrition in the paper even if the 1996 World Food Summit referred to nutritious foods and meeting dietary needs; they are not the same thing. F.ex. Abstract: “comprehensive nutrition model to investigate” “food and nutrition security”; also page 6: “Most research on food security has so far focused on energy consumption (calories). However, having sufficient energy does not guarantee a nutritionally adequate diet.”; page 29, line 3 “nutrition consumption”

The paper suggests that they developed a nutrition model. I did not see a nutrition model in the paper; instead the authors link food supply with nutrient content of such supply, which does not require a model. Models would consider a variety of factors affecting nutrition, such as income, female secondary education, a safe WASH environment, etc.

The paper should also compare the results with other models that modeled macronutrient and micronutrient adequacy, such as <https://www.nature.com/articles/s41893-018-0192-z>
As well as global models that only focus on modeling trade.

SPECIFICS

P2, line 45, framework consists of, maybe framework that consists of.

P3, line 8, rather than ref 3 for the number of chronically undernourished, please use one of the original sources, such as the SOFI report, also replace “today” with the year that is referenced. The values and definitions are in flux.

P3, line 21 “change the agricultural land use” or land use?

P3, line 28 “about 23% of the food produced was traded” – please add year as this changes (gradual increases), ideally referring to the source, which are FAOSTAT trade data.

P3, line 30, “which feeds 2-3 billion people” – suggest to add where they are located

Section 1.1

Please also refer to partial agricultural equilibrium models as well as Integrated Assessment models that often consider biophysical constraints through tight or loose linkages with other modeling tools.

Please describe how marketing margins, and trade restrictions that are already built into existing modeling frameworks address some of the concerns you raise in this section.

P.4, line 44, please explain how Brazil “secures land ownership in neighboring countries”. Does this refer to Uruguay, Paraguay? I expect readers would be interested in this.

P4, line 47, crises should be crisis.

P4, line 56, “models”... “fully understand” .. maybe “fully describe”?

P4, line 60 – my understanding is that market-equilibrium models can include non-traded goods;

P5, line 35, carries should be carry, who should be which or that

P5, line 35, “Neither does it” what does it refer to? Maybe add that there are numerous (countless?) regional (within and across country) trade modeling assessments for various purposes, many are focused on de factor and de jure regional trading blocs (Eastern African maize trade), MERCOSUR, etc. etc. plus multiple papers focusing on intra-national trade. These are not necessarily more recent studies. Many of them from the 1990s and even 1980s.

P5, line 46 “less developed”, maybe exchange with “lower-income”

P5, line 48, varies should be vary

P6, line 9: “over-consumption of unhealth food... obesity”, even over-consumption of healthy food can lead to obesity.

P6, line 14, “1996, the Food and Agriculture Organization (FAO) amplified the definition of food

security to include a sufficient supply of nutrients to provide a healthy diet." I don't think that FAO referred to a "healthy diet". Please review and reference.

P6, line 14, please mention that this is SDG 2

P6, line 23, references 45/46, suggest to add some global assessments of vegetables, animal source foods or similar, not 2 UK focused studies only

P6, line 35-37, please add a ref re "FAO's standard." And why is orange listed as a vegetable?

P7, line 26, "dietary of a country" lacks a noun

P7, line 33, "consumption of food", does this refer to food intake or availability for consumption?

P7, line 39, "unconsumed domestic supply", does this mean that stocks are not considered?

Tabl 1: "initial GDP". FAO does not develop such data, suggest to use the original source, ditto for initial population, not clear why these are listed as not being dynamic, typical diet is usually not the same as a country's food supply, how is "household waste" linked to "losses"? how is the percentage of loss derived from FBS?, re current nutrient supply consumed – suggest to clarify that this does not consider national inequities in access to food (that can be reflected to s

Page 9, line 23 "adjusted to represent to food", not clear

P9, line 32, diets are related to costs and income – here it seems that diets are derived from production (and imports) only? Has this been done before? Please add a reference. Also relates to the question of how population growth is considered. I also don't think that countries strive to reflect "typical diets". If that were the case, sorghum, barley and roots and tuber food availability levels would not be changing in Africa and per capita rice consumption would not be declining in Asia (and growing in Africa). Countries do not want to remain in a status quo!

Page 9, line 47: "The most commonly fortified food is wheat" – please add a reference for this statement, and also a reference for using USDA food composition tables (that do not do well in many low-income countries), but have been used in other studies.

Page 10, line 49, how do IIASA population projections link up with the UN data that are used for 2000?

Page 11, line 7, how is affordability defined? Based on cost of food data, like here:

<https://linkinghub.elsevier.com/retrieve/pii/S2214109X19304474>

Section 2.2.2 is very interesting but would benefit from references, for example, for the assumption that countries prefer to trade with richer countries.

Page 13, line 46, what is the trade limit for larger countries?

Page 16, Table 4, please update/correct the data sources for GDP and population – both show FAO which is not the custodian for either one.

A lot of the information in pages 17-19 could be put into an online appendix.

Page 22, line 27 – only some models aggregate countries. Equilibrium model, by design are equilibrating food supply and demand at national levels, so data are used for every single country – and each country is de facto also an agent.

Page 23 – nutrient intake data – is different from FAOSTAT food consumption data. I think this was clarified earlier on but went missing here.

Page 25, line 58: "nutrition consumption" please reword

Page 29, line 6-8 suggests that inequity matters for large countries; instead I would put forward that inequity matters for countries with high inequity in income and access to resources. Such countries can be small and large; inequity is not such large issue in China (albeit has been growing substantially). Inequity in food distribution is, f.ex. large in small countries, such as Chad or CAR, where a minority of the population has adequate or more than adequate access to food and nutrition while a majority has not.

Decision letter (RSOS-201587.R0)

Dear Dr Ge

On behalf of the Editors, we are pleased to inform you that your Manuscript RSOS-201587 "Food and Nutrition Security under Global Trade - A relation-driven agent-based global trade model" has been accepted for publication in Royal Society Open Science subject to minor revision in accordance with the referees' reports. Please find the referees' comments along with any feedback from the Editors below my signature.

(The comments from the referees are quite long and detailed, but their overall verdicts are positive and it appears that it will be straightforward for you to take account of their comments.)

Please submit your revised manuscript and required files (see below) no later than 7 days from today's (ie 18-Nov-2020) date. Note: the ScholarOne system will 'lock' if submission of the revision is attempted 7 or more days after the deadline. If you do not think you will be able to meet this deadline please contact the editorial office immediately.

on behalf of Prof Peter Haynes (Subject Editor)
openscience@royalsociety.org

Associate Editor Comments to Author:

Thank you for this submission which the reviewers are broadly in favour of accepting, but there are a range of modifications they suggest that would add value to your paper. Please carefully incorporate these changes and we'll look forward to receiving a revision from you in due course. Generally, the journal encourages authors to complete minor revisions within a week, but we

understand this is not a normal year, so if you need a minor extension, I'm sure the editorial office can assist.

Reviewer comments to Author:

Reviewer: 1

Comments to the Author(s)

I would like to congratulate the authors on their very interesting and well-written work. I have written down my suggestions, questions, and considerations as I was going through the text, which I hope will be found useful. That is, perhaps, not the best way to clarify what are the exact revisions I'd suggest the Editor to ask the authors to make. For clarity, I enumerate the specific actions needed below -- they refer to the comments (a) to (h) that follow.

1. Clarifications needed for comments (a) to (c) can be done on section 2.2.2. One way of doing it is suggested at the end of comment (c);
2. Regarding comment (d), a brief description of the structure of maximum amount of trade is needed. By structure I mean is it a constant universal value? Is it country specific? Food-category specific? The sensitivity analysis suggested is not needed if the authors provide further explanation of why it is not reasonable to expect this parameter to change significantly the behavior of the model;
3. To address my comment (e), clarification about the order by which food-categories are traded is needed. If the order is not changed/randomized between years, a few lines discussing why this wouldn't introduce a bias against the trade of the last categories, or why this is not important for the results and the objectives of the paper would be required;
4. Comments (f) to (h) require clarifications, as suggested in the respective comments;

In my understanding the work is scientifically sound and novel enough for publication. In its current writing the manuscript raises a few questions that need to be addressed in order to better reflect that. Including the additions in the four points above would make the scientific content in the manuscript described well-enough to be published.

Kind regards,
Alvaro

--- Comments ---

(a) page 11, paragraph starting at lines 33-34. It is not clear if the trade volume information is corrected for the first two elements (GDP, and proximity). The authors wrote that historic trade volume will reflect these non-GDP and non-geographic factors, but isn't it reasonable to assume that trade volume will also be affected by GDP and geographic factors? Assume a country A exports the same volume in a given commodity in 2000 to two different neighboring countries, B and C, both with similar distances, but B having a significantly larger GDP than C. Would this third element, trade relationship, be assigned the same value for that country exporting that commodity, or have different values taking in consideration the difference in GDP between the two trade partners? Shouldn't the trade-relationship element for the export of that commodity from A to C have a higher value than that from A to B in this case? If not corrected, then it would be important to discuss why this wouldn't introduce a significant double-counting type of bias in favor of larger GDP / shorter distances when combining this third element with the previous two.

(b) page 12, paragraph starting at lines 3-4. In here the reader is left assuming that the ranking and matching process is calibrated somehow to account for trade-offs between the variables mentioned in the paragraph (production and transport costs, willingness to pay, etc...). Does that process take into consideration the effects of the different variables differently? For example, up to which point (quantity?) production costs outweigh transport costs in the match-making process? A few sentences in here explaining how such calibration allows the model to reproduce the dynamics of price implicitly would make the argument in this paragraph more convincing. As it is, the argument sounds a bit incomplete: the model includes all the information in the price, but does it do so in a way that emulates the price dynamics sufficiently well?

(c) page 12, paragraph starting at lines 15-16. The calibration described here is a valid way of addressing the questions above in an empirical way. As a suggestion, the authors would be able to address both issues above in a relatively short way by adding a sentence at the end of the paragraph at page 11, line 6 stating that the four elements are calibrated with real data, and then a few sentences in each of the two paragraphs I've mentioned before stating how the calibration addresses the issues, namely of the possible double-count bias of the third element (they don't need to be orthogonal, as long as their parameters are calibrated to match observed data), and of the price dynamics (the behavior is calibrated to match the observations, and therefore it should work at least as well as having a price variable explicitly). Regardless of the way they choose to address these questions, I think it would be useful to have it in this section (2.2.2.).

(d) page 13, paragraph starting at line 31. The paragraph does a good job of explaining why the addition of the maximum amount of trade is important, but leaves the reader wondering how this parameter is computed, and what are the uncertainties in the model's behavior caused by its addition. If this is described elsewhere, it is worth pointing it out. If not, it would be important to describe the structure of this parameter (is it a flat value for all countries? does it change over time, or over commodities?), and also to provide some form of sensitivity analysis regarding its parameterization. What would be a reasonable variation of this maximum amount? What would be a good metric to check the impact of such variation? By how much would that metric change under that variation? Could the literature help ground any of these answers?

(e) page 13, paragraph starting at line 49. Is the order with which the food categories are traded the same every year? If so, wouldn't it introduce a bias against the consumption of the latter categories in the order, since possibly all the macro and micro nutrients needed by a country could be met by the trade and production of the first categories? Would it be possible to randomize the order the categories are traded from year to year in NetLogo (or perhaps feeding a new randomized order into NetLogo each year)? And if so, can it be used to show that following a fixed order yields no significant difference on the final result?

(f) page 14, paragraph starting at line 3. Although the use of the word parameter implies that trade saturation is a quantity that can be set in the model, it is only much later in the text (section 4.1) that this parameter is fixed to different values, and its impact discussed. From this paragraph alone it could also be the case that the authors let the model run until market clearance, saving snapshots of each food category's trade round at fixed saturation values. A few words in here clarifying that saturation is a parameter in the sense that it can be fixed by the model might help some readers understand it more concretely in here, and later help in the reading of section 4.1.

(g) page 18, paragraph starting at lines 44-45. The authors' approach to calibrating their model is sufficient for the scope of the current work. It would be interesting to explore in a future work how the calibrated parameters in table 5 would vary across countries, or at least regions. The significant differences between the mean values in table 18 suggests the general parameterization done presently works better for some regions than some others. Also, the use of two metrics for finding the Pareto tradeoff between them is a valid approach, but it doesn't guarantee that the

solutions are matching volume and partner together. The same goes for the validation. In the future the authors might want to consider a single metric defined by number of matches (right partner with a volume difference within the same 20% range) over the total amount of trades. This single metric would allow the authors to pick the best parameterization in the random set of 10,000, or to inform some sort of gradient-descent / markov-chain mechanism, which might even converge before 10,000 runs (particularly if the authors use their current calibration as starting point). For the present work, a few lines discussing how these two caveats, namely lack of parameterization per country, and the uncontrolled correlations between trade volumes and partners, are not an issue for its stated purposes (as in page 6, paragraph starting at line 29) would make section 3.1, and subsequently section 3.2, better.

(h) page 22, paragraph starting at line 55. If indeed trade saturation is a parameter that can be fixed in the model, the calibration and validation sections don't discussed explicitly to which value it was set for the analyses there. From my reading, currently the model doesn't have an endogenous mechanism that stops trading from happening before market clearance. I understand it is beyond the scope of the current work, but wouldn't it be reasonable to assume that relaxing the constraint of market clearance would allow the model to provide better predictions? If trade saturation is indeed a parameter, why not include it in the calibration process? In other words, it would be important to discuss precisely how trade saturation works in the model, and if it is indeed a fixed parameter, a constraint so to speak, why the choice of 1, or whatever choice was made in the calibration and validation phases, is better than, say 0.5, or a value that yielded the best calibration with the observed data?

Reviewer: 2

Comments to the Author(s)

Food and Nutrition Security under Global Trade

GENERAL

The paper provides a useful contribution to the current set of global trade models using countries as agents that trade based on 4 main factors or criteria that are identified based on analysis of historic data. Key challenges that the authors could address include, first, move most of the model development specificities into the appendix. The paper is very long and many readers will not be able to hang on for the ride; try to address, if feasible some of the already identified challenges, such as relating future trade only to history and not to changing trends in diets (linked empirically to changes in incomes and urbanization) in low-income countries, and in high-income countries, to reductions in per capita consumption of some food items, like milk, sugary drinks, and a few other commodities.

P 28, lines 26 notes that poor countries are penalized by trade. Other studies have found that poor countries always benefit nutritionally from trade, pretty much regardless of scenario.

The results on pages 23-25 could be tightened and presented differently. My black and white printout showed nothing, also which trade saturation picture comes closest to reality?

The paper should take a bit more care to adequately differentiate the terms food security and nutrition in the paper even if the 1996 World Food Summit referred to nutritious foods and meeting dietary needs; they are not the same thing. F.ex. Abstract: "comprehensive nutrition model to investigate" "food and nutrition security"; also page 6: "Most research on food security has so far focused on energy consumption (calories). However, having sufficient energy does not guarantee a nutritionally adequate diet."; page 29, line 3 "nutrition consumption"

The paper suggests that they developed a nutrition model. I did not see a nutrition model in the paper; instead the authors link food supply with nutrient content of such supply, which does not require a model. Models would consider a variety of factors affecting nutrition, such as income, female secondary education, a safe WASH environment, etc.

The paper should also compare the results with other models that modeled macronutrient and micronutrient adequacy, such as <https://www.nature.com/articles/s41893-018-0192-z>
As well as global models that only focus on modeling trade.

SPECIFICS

P2, line 45, framework consists of, maybe framework that consists of.

P3, line 8, rather than ref 3 for the number of chronically undernourished, please use one of the original sources, such as the SOFI report, also replace “today” with the year that is referenced. The values and definitions are in flux.

P3, line 21 “change the agricultural land use” or land use?

P3, line 28 “about 23% of the food produced was traded” – please add year as this changes (gradual increases), ideally referring to the source, which are FAOSTAT trade data.

P3, line 30, “which feeds 2-3 billion people” – suggest to add where they are located

Section 1.1

Please also refer to partial agricultural equilibrium models as well as Integrated Assessment models that often consider biophysical constraints through tight or loose linkages with other modeling tools.

Please describe how marketing margins, and trade restrictions that are already built into existing modeling frameworks address some of the concerns you raise in this section.

P.4, line 44, please explain how Brazil “secures land ownership in neighboring countries”. Does this refer to Uruguay, Paraguay? I expect readers would be interested in this.

P4, line 47, crises should be crisis.

P4, line 56, “models” ... “fully understand” .. maybe “fully describe”?

P4, line 60 – my understanding is that market-equilibrium models can include non-traded goods;

P5, line 35, carries should be carry, who should be which or that

P5, line 35, “Neither does it” what does it refer to? Maybe add that there are numerous (countless?) regional (within and across country) trade modeling assessments for various purposes, many are focused on de factor and de jure regional trading blocs (Eastern African maize trade), MERCOSUR, etc. etc. plus multiple papers focusing on intra-national trade. These are not necessarily more recent studies. Many of them from the 1990s and even 1980s.

P5, line 46 “less developed”, maybe exchange with “lower-income”

P5, line 48, varies should be vary

P6, line 9: “over-consumption of unhealth food... obesity”, even over-consumption of healthy food can lead to obesity.

P6, line 14, “1996, the Food and Agriculture Organization (FAO) amplified the definition of food security to include a sufficient supply of nutrients to provide a healthy diet.” I don’t think that FAO referred to a “healthy diet”. Please review and reference.

P6, line 14, please mention that this is SDG 2

P6, line 23, references 45/46, suggest to add some global assessments of vegetables, animal source foods or similar, not 2 UK focused studies only

P6, line 35-37, please add a ref re “FAO’s standard.” And why is orange listed as a vegetable?

P7, line 26, “dietary of a country” lacks a noun

P7, line 33, “consumption of food”, does this refer to food intake or availability for consumption?

P7, line 39, “unconsumed domestic supply”, does this mean that stocks are not considered?

Tabl 1: “initial GDP”. FAO does not develop such data, suggest to use the original source, ditto for initial population, not clear why these are listed as not being dynamic, typical diet is usually not the same as a country’s food supply, how is “household waste” linked to “losses”? how is the percentage of loss derived from FBS?, re current nutrient supply consumed – suggest to clarify that this does not consider national inequities in access to food (that can be reflected to

Page 9, line 23 “adjusted to represent to food”, not clear

P9, line 32, diets are related to costs and income – here it seems that diets are derived from production (and imports) only? Has this been done before? Please add a reference. Also relates to the question of how population growth is considered. I also don't think that countries strive to reflect "typical diets". If that were the case, sorghum, barley and roots and tuber food availability levels would not be changing in Africa and per capita rice consumption would not be declining in Asia (and growing in Africa). Countries do not want to remain in a status quo!

Page 9, line 47: "The most commonly fortified food is wheat" – please add a reference for this statement, and also a reference for using USDA food composition tables (that do not do well in many low-income countries), but have been used in other studies.

Page 10, line 49, how do IIASA population projections link up with the UN data that are used for 2000?

Page 11, line 7, how is affordability defined? Based on cost of food data, like here:

<https://linkinghub.elsevier.com/retrieve/pii/S2214109X19304474>

Section 2.2.2 is very interesting but would benefit from references, for example, for the assumption that countries prefer to trade with richer countries.

Page 13, line 46, what is the trade limit for larger countries?

Page 16, Table 4, please update/correct the data sources for GDP and population – both show FAO which is not the custodian for either one.

A lot of the information in pages 17-19 could be put into an online appendix.

Page 22, line 27 – only some models aggregate countries. Equilibrium model, by design are equilibrating food supply and demand at national levels, so data are used for every single country – and each country is de facto also an agent.

Page 23 – nutrient intake data – is different from FAOSTAT food consumption data. I think this was clarified earlier on but went missing here.

Page 25, line 58: "nutrition consumption" please reword

Page 29, line 6-8 suggests that inequity matters for large countries; instead I would put forward that inequity matters for countries with high inequity in income and access to resources. Such countries can be small and large; inequity is not such large issue in China (albeit has been growing substantially). Inequity in food distribution is, f.ex. large in small countries, such as Chad or CAR, where a minority of the population has adequate or more than adequate access to food and nutrition while a majority has not.

===PREPARING YOUR MANUSCRIPT===

Please ensure that you include an acknowledgements' section before your reference list/bibliography. This should acknowledge anyone who assisted with your work, but does not

qualify as an author per the guidelines at <https://royalsociety.org/journals/ethics-policies/openness/>.

===PREPARING YOUR REVISION IN SCHOLARONE===

- Ensure that your data access statement meets the requirements at <https://royalsociety.org/journals/authors/author-guidelines/#data>. You should ensure that you cite the dataset in your reference list. If you have deposited data etc in the Dryad repository, please only include the 'For publication' link at this stage. You should remove the 'For review' link.
- If you are requesting an article processing charge waiver, you must select the relevant waiver option (if requesting a discretionary waiver, the form should have been uploaded at Step 3 'File upload' above).
- If you have uploaded ESM files, please ensure you follow the guidance at <https://royalsociety.org/journals/authors/author-guidelines/#supplementary-material> to include a suitable title and informative caption. An example of appropriate titling and captioning may be found at https://figshare.com/articles/Table_S2_from_Is_there_a_trade-off_between_peak_performance_and_performance_breadth_across_temperatures_for_aerobic_sc_ope_in_teleost_fishes_/3843624.

Author's Response to Decision Letter for (RSOS-201587.R0)

See Appendix A.

Decision letter (RSOS-201587.R1)

Dear Dr Ge,

It is a pleasure to accept your manuscript entitled "Food and Nutrition Security under Global Trade - A relation-driven agent-based global trade model" in its current form for publication in Royal Society Open Science.

Due to rapid publication and an extremely tight schedule, if comments are not received, your paper may experience a delay in publication. Royal Society Open Science operates under a continuous publication model. Your article will be published straight into the next open issue and this will be the final version of the paper. As such, it can be cited immediately by other researchers. As the issue version of your paper will be the only version to be published I would

advise you to check your proofs thoroughly as changes cannot be made once the paper is published.

on behalf of Peter Haynes (Subject Editor)
openscience@royalsociety.org

Appendix A

First we would like to thank two anonymous reviewers for their constructive comments and suggestions, which help us greatly improve the quality and clarity of the paper. We address the comments of the reviewers as below.

Reviewer: 1

Comments to the Author(s)

I would like to congratulate the authors on their very interesting and well-written work. I have written down my suggestions, questions, and considerations as I was going through the text, which I hope will be found useful. That is, perhaps, not the best way to clarify what are the exact revisions I'd suggest the Editor to ask the authors to make. For clarity, I enumerate the specific actions needed below -- they refer to the comments (a) to (h) that follow.

1. Clarifications needed for comments (a) to (c) can be done on section 2.2.2. One way of doing it is suggested at the end of comment (c);

Please see point a-c below

2. Regarding comment (d), a brief description of the structure of maximum amount of trade is needed. By structure I mean is it a constant universal value? Is it country specific? Food-category specific? The sensitivity analysis suggested is not needed if the authors provide further explanation of why it is not reasonable to expect this parameter to change significantly the behavior of the model;

Please see point d below

3. To address my comment (e), clarification about the order by which food-categories are traded is needed. If the order is not changed/randomized between years, a few lines discussing why this wouldn't introduce a bias against the trade of the last categories, or why this is not important for the results and the objectives of the paper would be required;

Please see point e below

4. Comments (f) to (h) require clarifications, as suggested in the respective comments;

Please see point f-h below

In my understanding the work is scientifically sound and novel enough for publication. In its current writing the manuscript raises a few questions that need to be addressed in order to better reflect that. Including the additions in the four points above would make the scientific content in the manuscript described well-enough to be published.

Kind regards,
Alvaro

--- Comments ---

(a) page 11, paragraph starting at lines 33-34. It is not clear if the trade volume information is corrected for the first two elements (GDP, and proximity). The authors wrote that historic trade volume will reflect these non-GDP and non-geographic factors, but isn't it reasonable to assume that trade volume will also be affected by GDP and geographic factors? Assume a country A exports the

same volume in a given commodity in 2000 to two different neighboring countries, B and C, both with similar distances, but B having a significantly larger GDP than C. Would this third element, trade relationship, be assigned the same value for that country exporting that commodity, or have different values taking in consideration the difference in GDP between the two trade partners? Shouldn't the trade-relationship element for the export of that commodity from A to C have a higher value than that from A to B in this case? If not corrected, then it would be important to discuss why this wouldn't introduce a significant double-counting type of bias in favor of larger GDP / shorter distances when combining this third element with the previous two.

Although the four elements might be correlated (e.g. many countries that trade often are also geographically close), they do not coincide. For example, countries that have developed historical trade links may not be geographically close, such as the common wealth countries; and countries that are geographically close may not trade as much, such as between the U.S. and Cuba. Hence, we should still be able to distinguish the effect of the four elements from the data. We included the discussion in 2.2.2.

(b) page 12, paragraph starting at lines 3-4. In here the reader is left assuming that the ranking and matching process is calibrated somehow to account for trade-offs between the variables mentioned in the paragraph (production and transport costs, willingness to pay, etc...). Does that process take into consideration the effects of the different variables differently? For example, up to which point (quantity?) production costs outweigh transport costs in the match-making process? A few sentences in here explaining how such calibration allows the model to reproduce the dynamics of price implicitly would make the argument in this paragraph more convincing. As it is, the argument sounds a bit incomplete: the model includes all the information in the price, but does it do so in a way that emulates the price dynamics sufficiently well?

In this model we do not try to emulate price dynamics or predict future price. We are only saying that how the price mechanism allocated commodities among countries will be (partially) incorporated by accounting for countries' GDP per capita (as a proxy for ability to pay) and geographic distance (as a proxy for transport cost). We included the discussion in 2.2.2.

(c) page 12, paragraph starting at lines 15-16. The calibration described here is a valid way of addressing the questions above in an empirical way. As a suggestion, the authors would be able to address both issues above in a relatively short way by adding a sentence at the end of the paragraph at page 11, line 6 stating that the four elements are calibrated with real data, and then a few sentences in each of the two paragraphs I've mentioned before stating how the calibration addresses the issues, namely of the possible double-count bias of the third element (they don't need to be orthogonal, as long as their parameters are calibrated to match observed data), and of the price dynamics (the behavior is calibrated to match the observations, and therefore it should work at least as well as having a price variable explicitly). Regardless of the way they choose to address these questions, I think it would be useful to have it in this section (2.2.2.).

Yes we have included the clarification and discussion in 2.2.2.

(d) page 13, paragraph starting at line 31. The paragraph does a good job of explaining why the addition of the maximum amount of trade is important, but leaves the reader wondering how this parameter is computed, and what are the uncertainties in the model's behavior caused by its addition. If this is described elsewhere, it is worth pointing it out. If not, it would be important to describe the structure of this parameter (is it a flat value for all countries? does it change over time,

or over commodities?), and also to provide some form of sensitivity analysis regarding its parameterization. What would be a reasonable variation of this maximum amount? What would be a good metric to check the impact of such variation? By how much would that metric change under that variation? Could the literature help ground any of these answers?

The maximum amount depends on the commodity, and is set to be the total volume available for trade divided by a constant, which is set to be 500. We did a sensitivity analysis and change the constant from 100 to 1000, and get very similar results. We want to emphasise that this is not the maximum amount of commodity a country can trade in total, it is the maximum amount a country can trade *in one round*, and trades can take several rounds (on average between 5-20) to finish. Therefore we do not expect the results to be sensitive to the limit we set. We have clarified it in the text.

(e) page 13, paragraph starting at line 49. Is the order with which the food categories are traded the same every year? If so, wouldn't it introduce a bias against the consumption of the latter categories in the order, since possibly all the macro and micro nutrients needed by a country could be met by the trade and production of the first categories? Would it be possible to randomize the order the categories are traded from year to year in NetLogo (or perhaps feeding a new randomized order into NetLogo each year)? And if so, can it be used to show that following a fixed order yields no significant difference on the final result?

The order is the same, however, it should not cause any bias to the results, because the trade of each commodity is independent of each other, and the countries are not trying to optimise their nutrient intake. Say, the countries will start from wheat, and then rice, and then vegetables, and meat, and fish etc (there are 91 food commodities that they trade), and each trade decision is made independent of trade decisions of other foods.

If they are trying to optimise nutrient intake, then order of trade will matter. In this version we do not consider that. In a later version of the model as one of the scenarios we introduce an 'ideal diet' for each country where they try to meet nutrient requirement with minimum change to their current diet (to respect tradition and culture) using linear programming.

(f) page 14, paragraph starting at line 3. Although the use of the word parameter implies that trade saturation is a quantity that can be set in the model, it is only much later in the text (section 4.1) that this parameter is fixed to different values, and its impact discussed. From this paragraph alone it could also be the case that the authors let the model run until market clearance, saving snapshots of each food category's trade round at fixed saturation values. A few words in here clarifying that saturation is a parameter in the sense that it can be fixed by the model might help some readers understand it more concretely in here, and later help in the reading of section 4.1.

We have clarified the issue and change the wording.

(g) page 18, paragraph starting at lines 44-45. The authors' approach to calibrating their model is sufficient for the scope of the current work. It would be interesting to explore in a future work how the calibrated parameters in table 5 would vary across countries, or at least regions. The significant differences between the mean values in table 18 suggests the general parameterization done presently works better for some regions than some others. Also, the use of two metrics for finding the Pareto tradeoff between them is a valid approach, but it doesn't guarantee that the solutions are matching volume and partner together. The same goes for the validation. In the future the authors might want to consider a single metric defined by number of matches (right partner with a volume

difference within the same 20% range) over the total amount of trades. This single metric would allow the authors to pick the best parameterization in the random set of 10,000, or to inform some sort of gradient-descent / markov-chain mechanism, which might even converge before 10,000 runs (particularly if the authors use their current calibration as starting point). For the present work, a few lines discussing how these two caveats, namely lack of parameterization per country, and the uncontrolled correlations between trade volumes and partners, are not an issue for its stated purposes (as in page 6, paragraph starting at line 29) would make section 3.1, and subsequently section 3.2, better.

We have included the discussion on how the lack of parameterization per country and the uncontrolled correlations between trade volumes and partners are not an issue for the purposes of the study.

(h) page 22, paragraph starting at line 55. If indeed trade saturation is a parameter that can be fixed in the model, the calibration and validation sections don't discussed explicitly to which value it was set for the analyses there. From my reading, currently the model doesn't have an endogenous mechanism that stops trading from happening before market clearance. I understand it is beyond the scope of the current work, but wouldn't it be reasonable to assume that relaxing the constraint of market clearance would allow the model to provide better predictions? If trade saturation is indeed a parameter, why not include it in the calibration process? In other words, it would be important to discuss precisely how trade saturation works in the model, and if it is indeed a fixed parameter, a constraint so to speak, why the choice of 1, or whatever choice was made in the calibration and validation phases, is better than, say 0.5, or a value that yielded the best calibration with the observed data?

'Trade saturation' is an intervention parameter that we vary in the computer experiment. It is not fixed, and is exogenous to the model. We agree with the reviewer that one way to approach it is to calibrate it with data, and find the optimal value for it that best fit the data. However, unlike the four elements w.r.t trade priority, which are intrinsic values in the model to be calibrated, we believe trade saturation is an intervention parameter, which trade policies and global geopolitics will change. We want to be able to alter its value to see its impact on results. We have included the clarification in section 4.1.

Reviewer: 2

Comments to the Author(s)

Food and Nutrition Security under Global Trade

GENERAL

The paper provides a useful contribution to the current set of global trade models using countries as agents that trade based on 4 main factors or criteria that are identified based on analysis of historic data. Key challenges that the authors could address include, first, move most of the model development specificities into the appendix. The paper is very long and many readers will not be able to hang on for the ride; try to address, if feasible some of the already identified challenges, such as relating future trade only to history and not to changing trends in diets (linked empirically to changes in incomes and urbanization) in low-income countries, and in high-income countries, to reductions in per capita consumption of some food items, like milk, sugary drinks, and a few other commodities.

We have addressed the above comments, see 'SPECIFICS'.

P 28, lines 26 notes that poor countries are penalized by trade. Other studies have found that poor countries always benefit nutritionally from trade, pretty much regardless of scenario. The results on pages 23-25 could be tightened and presented differently. My black and white printout showed nothing, also which trade saturation picture comes closest to reality? The paper should take a bit more care to adequately differentiate the terms food security and nutrition in the paper even if the 1996 World Food Summit referred to nutritious foods and meeting dietary needs; they are not the same thing. F.ex. Abstract: "comprehensive nutrition model to investigate" "food and nutrition security"; also page 6: "Most research on food security has so far focused on energy consumption (calories). However, having sufficient energy does not guarantee a nutritionally adequate diet."; page 29, line 3 "nutrition consumption"

We have addressed the above comments, see 'SPECIFICS'.

The paper suggests that they developed a nutrition model. I did not see a nutrition model in the paper; instead the authors link food supply with nutrient content of such supply, which does not require a model. Models would consider a variety of factors affecting nutrition, such as income, female secondary education, a safe WASH environment, etc.

We have correct the wording, instead of a nutrition model, we link the trade model to the nutrition formula table based on food consumption that colleagues previously developed. We have corrected the phrase we use throughout the paper.

The paper should also compare the results with other models that modeled macronutrient and micronutrient adequacy, such as <https://www.nature.com/articles/s41893-018-0192-z> As well as global models that only focus on modeling trade.

We have included the comparison of results in this study with previous ones in the discussion section.

SPECIFICS

P2, line 45, framework consists of, maybe framework that consists of.

Done

P3, line 8, rather than ref 3 for the number of chronically undernourished, please use one of the original sources, such as the SOFI report, also replace "today" with the year that is referenced. The values and definitions are in flux.

Done

P3, line 21 "change the agricultural land use" or land use?

Done

P3, line 28 "about 23% of the food produced was traded"—please add year as this changes (gradual increases), ideally referring to the source, which are FAOSTAT trade data.

Done

P3, line 30, "which feeds 2-3 billion people"—suggest to add where they are located

Done

Section 1.1

Please also refer to partial agricultural equilibrium models as well as Integrated Assessment models that often consider biophysical constraints through tight or loose linkages with other modeling tools. Please describe how marketing margins, and trade restrictions that are already built into existing modeling frameworks address some of the concerns you raise in this section.

We include partial agricultural equilibrium models such as CAPRI in section 1.1. The main issue with PE or GE models both are built upon the key assumptions of equilibrium, price elasticities and the market-clearing condition. In this study we want to relax these assumption. Moreover, these theoretical framework offers limited possibilities for rigorous testing against historical data and experience. In other words, the underlying assumptions of these framework are empirically grounded. We have discussed these issues in 1.1.

P.4, line 44, please explain how Brazil “secures land ownership in neighboring countries”. Does this refer to Uruguay, Paraguay? I expect readers would be interested in this.

Paraguay and Bolivia, included in the text

P4, line 47, crises should be crisis.

Done

P4, line 56, “models”... “fully understand”.. maybe “fully describe”?

Done

P4, line 60—my understanding is that market-equilibrium models can include non-traded goods;

We have removed that statement

P5, line 35, carries should be carry, who should be which or that

Done

P5, line 35, “Neither does it” what does it refer to? Maybe add that there are numerous (countless?) regional (within and across country) trade modeling assessments for various purposes, many are focused on de facto and de jure regional trading blocs (Eastern African maize trade), MERCOSUR, etc. etc. plus multiple papers focusing on intra-national trade. These are not necessarily more recent studies. Many of them from the 1990s and even 1980s.

Done

P5, line 46 “less developed”, maybe exchange with “lower-income”

Done

P5, line 48, varies should be vary

Done

P6, line 9: “over-consumption of unhealth food... obesity”, even over-consumption of healthy food can lead to obesity.

We have rewritten the sentence

P6, line 14, “1996, the Food and Agriculture Organization (FAO) amplified the definition of food security to include a sufficient supply of nutrients to provide a healthy diet.” I don’t think that FAO referred to a “healthy diet”. Please review and reference.

Done

P6, line 14, please mention that this is SDG 2

Done

P6, line 23, references 45/46, suggest to add some global assessments of vegetables, animal source foods or similar, not 2 UK focused studies only

Done

P6, line 35-37, please add a ref re “FAO’s standard.” And why is orange listed as a vegetable?

Corrected

P7, line 26, “dietary of a country” lacks a noun

Corrected

P7, line 33, “consumption of food”, does this refer to food intake or availability for consumption?

Food intake, corrected

P7, line 39, “unconsumed domestic supply”, does this mean that stocks are not considered?

Tabl 1: “initial GDP”. FAO does not develop such data, suggest to use the original source, ditto for initial population, not clear why these are listed as not being dynamic, typical diet is usually not the same as a country’s food supply, how is “household waste” linked to “losses”? how is the percentage of loss derived from FBS?, re current nutrient supply consumed—suggest to clarify that this does not consider national inequities in access to food (that can be reflected to s

The annual production is dynamic, the *initial* production, which is an average of production between 2000 and 2002 is not dynamic. We have also clarified that here we do not consider national inequalities in access to food.

Page 9, line 23 “adjusted to represent to food”, not clear

Corrected: “adjusted by the proportion that is not edible (e.g. banana peels) and wasted”

P9, line 32, diets are related to costs and income—here it seems that diets are derived from production (and imports) only? Has this been done before? Please add a reference. Also relates to the question of how population growth is considered. I also don’t think that countries strive to reflect “typical diets”. If that were the case, sorghum, barley and roots and tuber food availability levels would not be changing in Africa and per capita rice consumption would not be declining in Asia (and growing in Africa). Countries do not want to remain in a status quo!

Typical diet in 2000 is only used as a baseline, *the demand for food each year does change!* Changes in food demand is assumed to be proportional to changes in the aggregate production, to reflect the fact that the diet of a people only changes gradually (not drastically) overtime. We have clarified that in the text.

Page 9, line 47: “The most commonly fortified food is wheat”—please add a reference for this statement, and also a reference for using USDA food composition tables (that do not do well in many low-income countries), but have been used in other studies.

Reference added

Page 10, line 49, how do IIASA population projections link up with the UN data that are used for 2000?

We use UN data for population figures in the past from 2000 to 2013, and IIASA projections for population data in the future until 2050. We actually did not use the projected results till 2050 in the paper (which will be included in another paper about scenario analysis), so we removed the section about IIASA projections.

Page 11, line 7, how is affordability defined? Based on cost of food data, like here:

<https://linkinghub.elsevier.com/retrieve/pii/S2214109X19304474>

We have changed the wording. What we really mean here is countries will consider a trading partner's 'ability to pay', for which they use GDP per capita as a proxy, when they choose trade partners.

Section 2.2.2 is very interesting but would benefit from references, for example, for the assumption that countries prefer to trade with richer countries.

We have not found any references that says countries prefer to trade with richer partners than poorer ones. However, here GDP per capita serves as a proxy for affordability and ability to pay, and we have changed the wording accordingly.

Page 13, line 46, what is the trade limit for larger countries?

It depends on what commodity. The limit is set as the total volume available for export divided by a constant, which is set to be 500 in the model.

Page 16, Table 4, please update/correct the data sources for GDP and population—both show FAO which is not the custodian for either one.

Done

A lot of the information in pages 17-19 could be put into an online appendix.

We have decided to keep the calibration section in the main text as we think it is an important and integrated part of the analysis, and move the calibration results to the appendix.

Page 22, line 27—only some models aggregate countries. Equilibrium model, by design are equilibrating food supply and demand at national levels, so data are used for every single country—and each country is de facto also an agent.

Corrected

Page 23—nutrient intake data—is different from FAOSTAT food consumption data. I think this was clarified earlier on but went missing here.

Clarified

Page 25, line 58: "nutrition consumption" please reword

Done

Page 29, line 6-8 suggests that inequity matters for large countries; instead I would put forward that inequity matters for countries with high inequity in income and access to resources. Such countries can be small and large; inequity is not such large issue in China (albeit has been growing substantially). Inequity in food distribution is, f.ex. large in small countries, such as Chad or CAR, where a minority of the population has adequate or more than adequate access to food and nutrition while a majority has not.

We have modified the text to reflect the comment (removing comments about large countries, and discuss countries that has big internal inequality).